# SeRI: Gradient-Free Sensitive Region Identification in Decision-Based Black-Box Attacks

**Feiyang Wang**[1,2]**, Xingquan Zuo**[1,2,†]**, Hai Huang**[1,2]**, Gang Chen**[3]**, Hangwei Qian**[4]

[1]Beijing University of Posts and Telecommunications, China
[2]Key Laboratory of Trustworthy Distributed Computing and Services, China
[3]Victoria University of Wellington, New Zealand
[4]CFAR and IHPC, Agency for Science, Technology and Research (A*STAR), Singapore
[†]Corresponding author: `zuoxq@bupt.edu.cn`

## Abstract

Deep neural networks (DNNs) are highly vulnerable to adversarial attacks, where small, carefully crafted perturbations are added to input images to cause misclassification. These perturbations are particularly effective when concentrated in *sensitive regions* of an image. However, in decision-based black-box settings, where only the top-1 predicted label is observable and query budgets are strictly limited, identifying sensitive regions becomes extremely challenging. This issue is critical because without accurate region information, decision-based attacks cannot refine adversarial examples effectively, limiting both their efficiency and accuracy. We propose *Sensitive Region Identification* (**SeRI**), the first decision-based method that assigns a continuous sensitivity score to each image pixel. It enables fine-grained region discovery and substantially improves the efficiency of adversarial attacks, all without access to gradients, confidence scores, or surrogate models. SeRI progressively partitions the image into finer sub-regions and refines a continuous sensitivity score to capture their true importance. At each iteration, it generates two perturbation variants of the selected region by scaling its magnitude up or down, and compares their decision boundaries to derive an accurate, continuous characterization of pixel sensitivity. SeRI further divides selected region into smaller sub-regions, recursively refining the search for sensitive areas. This recursive refinement process enables more precise sensitivity estimation through fine-grained analysis, distinguishing SeRI from prior binary or one-shot region selection approaches. Experiments on two benchmark datasets show that SeRI significantly enhances state-of-the-art decision-based attacks in both targeted and non-targeted attack scenarios. Moreover, SeRI produces precise heatmaps of sensitive image regions, providing strong validation of the attack process. The code is available at `https://github.com/BUPTAIOC/SeRI`.

## 1 Introduction

Deep neural networks (DNNs) have achieved superiority in tasks such as image classification Brunner et al. (2019). However, they remain highly susceptible to carefully crafted adversarial examples generated by adversarial attacks Dong et al. (2020); Chen & Gu (2020); Chen et al. (2020a). Since these vulnerabilities undermine the reliability and security of machine learning systems, adversarial robustness has emerged as a central research focus. Li et al. (2021).

Adversarial attacks are typically grouped into **white-box** attacks Goodfellow et al. (2015); Carlini & Wagner (2017); Madry et al. (2018), **gray-box** attacks (or soft-label attacks, score-based attacks, confidence-based attacks) Chen et al. (2017); Ilyas et al. (2019), and **black-box** attacks Brendel et al. (2018); Cheng et al. (2019); Chen & Gu (2020), based on how much information the attacker can obtain about the target models. White-box and gray-box attacks require full or partial access to the target model, such as its network architectures, parameters, or confidence scores, which is rarely available in real-world scenarios Chen & Gu (2020). Consequently, black-box attacks have drawn significant interest in the research community Cai et al. (2022).

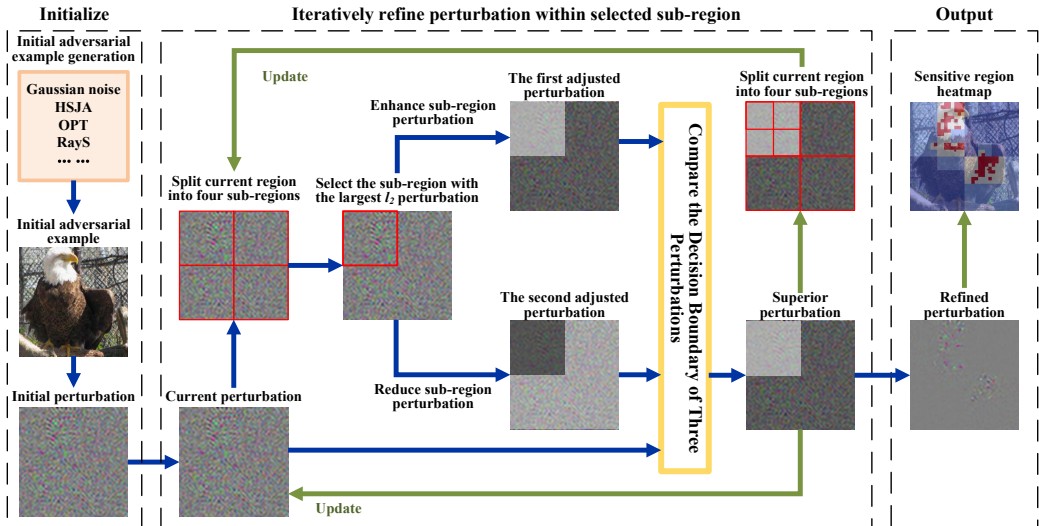

Figure 1: **Perturbation optimization process of SeRI.** The process starts by generating an initial adversarial example using Gaussian noise or a classical decision-based attack such as HSJA. The perturbation is split into four sub-regions, and the one with the highest $\ell_2$-norm is selected. Two new perturbations are created by enhancing or reducing the perturbation in that sub-region. SeRI compares their decision boundaries and keeps the one with the lowest value. In the next iteration, the selected sub-region is further subdivided for fine-grained optimization. A heatmap highlighting sensitive regions is then generated from the refined perturbation.

Black-box attacks fall into two main categories: **transfer-based** Wang et al. (2024); Sun et al. (2024); Park et al. (2024) and **decision-based** Chen & Gu (2020); Chen et al. (2020a); Reza et al. (2023); Wang et al. (2025). Transfer-based attacks train a surrogate model using the target model's data and then generate adversarial examples via white-box attack techniques. However, poor perturbation transferability often limits their attack success rate Reza et al. (2023). Decision-based attacks Brendel et al. (2018); Chen et al. (2020a); Chen & Gu (2020); Reza et al. (2023); Wang et al. (2025) aim to mislead a target DNN model by introducing minimal perturbations while operating under a limited query budget Li et al. (2021). Unlike white-box, gray-box, or transfer-based attacks, they do not rely on the target model's training data, architecture, or output confidence scores. Instead, they operate purely on the model's top-1 predicted label, which is often available in real-world systems Dong et al. (2019); Brunner et al. (2019).

**The role of sensitive regions in adversarial attacks**. It is well known that an image typically consists of both **sensitive regions** (i.e., salient objects like an eagle's head in Figure 2-(b)) and **non-sensitive regions** (i.e., irrelevant backgrounds in Figure 2-(b)). Focusing perturbations on sensitive regions, which are areas that contribute mostly to the model's prediction, has been shown to significantly improve the success rate of adversarial attacks. However, due to the limited information access and strict query constrains in black-box attack settings, identifying sensitive regions becomes a significant challenge Shi et al. (2022); Lin et al. (2023). Existing black-box attacks generally adopt two primary strategies to exploit sensitive regions.

The **first strategy** gives rise to the **transfer-based region-aware approaches** Chen et al. (2020b); Dong et al. (2020); Lovisotto et al. (2022); Lin et al. (2023). These methods generate sensitive region heatmaps from a surrogate model using white-box interpretability techniques, and then use the heatmaps to guide perturbation generation on the target model. However, models with different architectures, such as Vision Transformers Dosovitskiy (2020) and ResNets He et al. (2016), often focus on distinct features or regions Shi et al. (2022). As a result, the heatmaps generated from the surrogate model may fail to capture the decision-critical regions of the target model, leading to suboptimal attack performance.

The **second strategy** relies on **decision-based region sensitivity estimation**, where sensitivity is inferred from the model's top-1 prediction. PAR Shi et al. (2022) is the primary method that ex-

emplifies this approach with a patch-wise removal strategy: it deletes a perturbation block, queries the model, and labels the region as sensitive or not based on the hard-label prediction. To reduce query cost, PAR further uses a binary decision process that either keeps or removes the perturbation in a region, without any fine-grained adjustment. This approach restricts exploration of the solution space, leading to lower attack success rates. In reality, different pixels respond to perturbations with varying levels of sensitivity. This means *perturbations should be weighted in proportion to each pixel's sensitivity*, rather than applied in an all-or-nothing manner where an entire region is either kept or removed.

We introduce a **new definition of region sensitivity** grounded in the perturbation decision boundary (see Section 4 for details). Unlike prior binary formulations, this new definition enables a continuous and fine-grained quantification of sensitivity, providing a principled foundation for adaptive perturbation refinement in decision-based attacks. Different from PAR that treats each region as either sensitive or insensitive in PAR, our approach introduces an innovative *continuous sensitivity formulation* that assigns a perturbation weight score to each pixel. It enables *fine-grained control* over perturbation strength across regions, leading to targeted and effective perturbation optimization.

Building on this definition, we propose *Sensitive Region Identification* (SeRI), an approach that is efficient in both computation and queries, to adaptively optimize perturbations down to the level of individual pixels. SeRI is simple to implement in decision-based attack settings and can be seamlessly integrated as a plug-in perturbation optimizer to refine the perturbations generated by various baseline attackers like CGBA Reza et al. (2023). This "Attacker + SeRI" framework significantly enhances baseline attacker with minimal query-budget.

An overview of our approach is illustrated in Figure 1. Specifically, we generate two perturbation variants by either increasing or decreasing the perturbation strength within the selected region. These two variants, along with the original perturbation, are then compared based on their decision boundaries. For this comparison, we adopt the Approximation Decision Boundary Approach (ADBA) Wang et al. (2025), which operates effectively with a minimal query budget. The current perturbation is subsequently updated with the variant that yields the smallest estimated decision boundary. Experimental results on two datasets and three models confirm its effectiveness

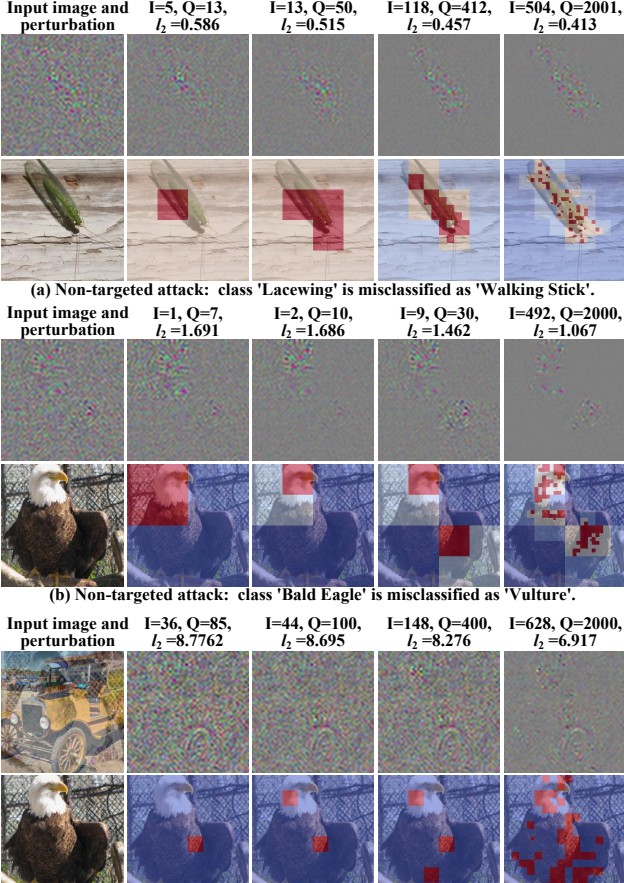

(a) Non-targeted attack: class 'Lacewing' is misclassified as 'Walking Stick'.

(b) Non-targeted attack: class 'Bald Eagle' is misclassified as 'Vulture'.

(c) Targeted attack: class 'Bald Eagle' is misclassified as 'Minivan'.

Figure 2: Heatmaps generated by SeRI, showing how the heatmaps, perturbations, and their $\ell_2$-norms evolve over iterations (I) and queries (Q).

and generalizability. The sensitivity-region heatmaps and the corresponding optimized perturbations produced by SeRI are shown in Figure 2. SeRI's precise thermal grading not only accelerates perturbation optimization by focusing on the most influential regions, but also produces heatmaps that align with human visual perception, thereby enhancing the interpretability of the attack process.

## 2 RELATED WORK

**Decision-based attacks** represent one of the most challenging setting in adversarial robustness, as only the model's top-1 predicted label is observable and no gradient information is available Chen & Gu (2020). Boundary AttackBrendel et al. (2018), Biased Boundary Attack Brunner et al. (2019) and AHA Li et al. (2021) perform random walks along the decision boundary to gradually reduce the perturbation strength. Triangle Attack Wang et al. (2022) designs a structured triangle-based perturbation in the low-frequency domain to improve efficiency. SurFree Maho et al. (2021) explores multiple carefully chosen perturbation directions simultaneously, without relying on gradient estimation. OPT Cheng et al. (2019) and Sign-OPT Cheng et al. (2020) reformulate the decision-based attacks as continuous optimization problem solvable via zeroth-order methods. RayS Chen & Gu (2020) and ADBA Wang et al. (2025) use a progressive search strategy that adaptively subdivides perturbation directions to accelerate convergence. Some attacks guide perturbation optimization by estimating the normal vector of the decision boundary at a boundary point. For example, qFool Liu et al. (2019) and GeoDA Rahmati et al. (2020) improve gradient estimation efficiency by leveraging the fact that decision boundaries near adversarial examples typically have low curvature. QEBA Li et al. (2020) reduces query complexity by performing subspace optimization in both the spatial and the frequency domains. HSJA Chen et al. (2020a), TA Ma et al. (2021), and BounceAttack Wan et al. (2024) estimates normal vectors to generate adversarial example. Achieving state-of-the-art performance under the $\ell_2$-norm, CGBA Reza et al. (2023) introduces a novel semicircular search strategy within a two-dimensional subspace to effectively navigate geometric complexities.

Although recent decision-based attacks achieve good performance, they usually ignore which region of the image are more sensitive to the model. Because they generate global perturbations without considering regional differences, the perturbations are often large and easy to notice.

**Sensitive region-based attacks**. Many classical XAI methods such as Grad-CAM Selvaraju et al. (2017), Occlusion Zeiler & Fergus (2014), LIME Ribeiro et al. (2016), and SHAP Lundberg & Lee (2017) can highlight important or influential regions of an input image. However, these methods are primarily designed to explain the prediction behavior of a model rather than to identify regions that are most useful for generating adversarial perturbations. In addition, they rely on gradients, confidence scores, or internal activations to compute importance, all of which are unavailable in hard-label decision-based attacks. Nonetheless, these interpretability techniques provide a natural motivation for using sensitive regions to guide perturbation generation in adversarial attacks.

Building on this idea, existing work attempts to explicitly use sensitive regions to improve attack quality. These methods can be broadly categorized into three groups: (1) attention-based methods Dong et al. (2020); Chen et al. (2020b), (2) surrogate-based methods Lin et al. (2023) and (3) decision-based methods Shi et al. (2022); Tao et al. (2023).

Superpixel-guided Attentional Attack (SGA) Dong et al. (2020) perturbs only the regions selected by attention maps. Attack on Attention (AoA) Chen et al. (2020b) optimizes an attention-based loss to improve transferability. However, both SGA and AoA need attention maps or activation features, which require white-box access and cannot be used in hard-label black-box settings. To avoid these requirements, SRA Lin et al. (2023) uses a surrogate model to generate sensitivity maps, but it needs the target dataset to be known. SaliencyAttack Dai et al. (2023) uses model-agnostic saliency detection, but the saliency maps may be inaccurate and fail to locate the truly important regions. Decision-based attacks such as HardBeat Tao et al. (2023) search for one vulnerable patch and perturb only that patch. However, when multiple regions jointly affect the classifier, focusing on only one patch becomes incomplete. PAR Shi et al. (2022) estimates region sensitivity through a patch-wise removal strategy, but its binary keep/remove decision is coarse and cannot reflect continuous importance inside each region. Overall, current methods cannot provide *continuous* region sensitivity under the strict decision-based setting.

## 3 PROBLEM DEFINITION

Let $p \in [0, 1]$ be an image pixel and $x = p^{C \times W \times H}$ represent a source image with channels $C$, width $W$, and height $H$, respectively, and $y(x)$ denote the true label of $x$. Let $f : x \to \{1, \dots, K\}$ represent a $K$-class image classification model. Given a source image $x$ which is classified correctly by model $f$ (i.e., $f(x) = y(x)$). The goal of a *decision-based black-box attacker* is to find an

adversarial example $\tilde{x} = \tilde{p}^{C \times W \times H}, \tilde{p} \in [0, 1]$, such that $f(\tilde{x}) \neq y(x)$ for *non-targeted attacks*, or $f(\tilde{x}) = f(x_{\text{tar}})$ for *targeted attacks*, while minimizing the perturbation strength $\|\tilde{x} - x\|_2$. Here, $x_{\text{tar}}$ is a given target image and $f(x_{\text{tar}}) \neq y(x)$, and $\| \cdot \|_2$ stands for the $\ell_2$-norm used to measure the perturbation strength. The problem of optimizing the adversarial example $\tilde{x}$ can be formulated as:

$$\arg \min_{\tilde{x}} \|\tilde{x} - x\|_2 \quad \text{s.t. } I(\tilde{x}) = 1, \tag{1}$$

where $I(\cdot)$ is an indicator function that determines whether the adversarial example $\tilde{x}$ is in the adversarial regions. For a non-targeted attack:

$$I(\tilde{x}) = \begin{cases} 1, & \text{if } f(\tilde{x}) \neq y(x), \\ -1, & \text{otherwise.} \end{cases} \tag{2}$$

For a targeted attack with a targeted image $x_{\text{target}}$:

$$I(\tilde{x}) = \begin{cases} 1, & \text{if } f(\tilde{x}) = f(x_{\text{target}}), \\ -1, & \text{otherwise.} \end{cases} \tag{3}$$

Let a perturbation be denoted as $d = v^{C \times W \times H}$, where $v \in [-1, 1]$ . An adversarial example $\tilde{x}$ is then given by $\tilde{x} = clamp(x + (\tilde{x} - x)) = clamp(x + d) = clamp(x + \|d\|_2 \cdot \frac{d}{\|d\|_2})$, where $clamp(\cdot)$ constrains each image pixel to the range $[0, 1]$. We define the decision boundary for any perturbation $d$ as $g(d) = \min\{r > 0 : I(x + r \cdot \frac{d}{\|d\|_2}) = 1\}$. Consequently, the optimization problem in Eq. 1 can be reformulated as:

$$\arg \min_{d} g(d) \quad \text{s.t. } I(x + g(d) \cdot \frac{d}{\|d\|_2}) = 1. \tag{4}$$

## 4 PROPOSED APPROACH

The proposed SeRI approach is applied after a base attacker (e.g., HSJA), refining its perturbation by identifying sensitive regions through additional queries. With total query budget $Q$ and SeRI query budget fraction $P$, query allocation is $(1 - P) \cdot Q$ for the base attacker and $P \cdot Q$ for SeRI. In this paper, we set $P = 20\%$ for all datasets and models, based on the parameter sensitivity analysis presented in Appendix C of the supplementary material. The introduction of base attackers are introduced in Section 2. Here, we focus on our proposed SeRI approach.

### 4.1 DEFINITION OF PERTURBATION SENSITIVITY

Existing sensitivity definitions can be roughly grouped into two families: gradient-based definitions and perturbation-based definitions. **Gradient-based definitions**, such Integrated Gradients (IG) Sundararajan et al. (2017), Grad-CAM Selvaraju et al. (2017), and their variants, define the sensitivity of a pixel by the gradient of a target score with respect to the input. Let $x \in [0, 1]^{C \times W \times H}$ denote the input image, $y(x)$ its true label, and $f_{y(x)}(x)$ the logit (or score) for class $y(x)$. A typical pixel-wise sensitivity can be written as

$$s_{c,w,h} = \left| \frac{\partial f_{y(x)}(x)}{\partial x_{c,w,h}} \right|, \qquad s_{w,h} = \sum_{c=1}^{C} \left| \frac{\partial f_{y(x)}(x)}{\partial x_{c,w,h}} \right|. \tag{5}$$

Such definitions capture which pixels or regions are most influential for the model's internal decision. However, they are not directly usable in decision-based black-box attacks, where gradients with respect to the input are completely unavailable.

**Perturbation-based definitions** estimate sensitivity by actively modifying the input and observing how the model output changes. Representative examples include Occlusion-Based explanations Zeiler & Fergus (2014), SRA Lin et al. (2023), and PAR Shi et al. (2022). Occlusion and SRA define pixel-wise sensitivity via the drop in the predicted confidence of the true class after perturbing a single pixel. Let $p_{y(x)}(x)$ denote the predicted probability of class $y(x)$. For each spatial location $(w, h)$, they consider a small $\ell_\infty$ perturbation of size $t > 0$ on all channels at that pixel:

$$s_{w,h} = \frac{p_{y(x)}(x) - p_{y(x)}(x + t\, \mathbf{e}_{w,h})}{t}, \tag{6}$$

where $\mathbf{e}_{w,h} \in \mathbb{R}^{C \times W \times H}$ is zero everywhere except at pixel $(w, h)$, with all $C$ channels equal to 1. Intuitively, $s_{w,h}$ measures how much the true-class probability decreases when only pixel $(w, h)$ is slightly perturbed, and thus serves as a pixel-wise sensitivity score in the black-box setting.

PAR, on the other hand, defines sensitivity at the patch level. Let $\tilde{x}$ be an adversarial example for $x$, and $d = \tilde{x} - x$ the corresponding perturbation. For a region (patch) $b$ with binary mask $M_b \in \{0, 1\}^{C \times W \times H}$, PAR uniformly rescales the perturbation inside $b$ by a factor $\kappa \in [0, 1]$, and the region-wise sensitivity of $b$ is then defined as:

$$S_{\text{PAR}}(b) = \min \left\{ \kappa \in [0, 1] \; : \; I\big(x + d^{(b,\kappa)}\big) = 1 \right\}, \qquad d^{(b,\kappa)} = d - (1 - \kappa)\, d \odot M_b, \quad (7)$$

where $\odot$ denotes element-wise multiplication and $I(\cdot)$ is an indicator function. A smaller value of $S_{\text{PAR}}(b)$ means that more perturbation in region $b$ can be removed while keeping the perturbation adversarial, hence $b$ is less sensitive; a larger value indicates a more sensitive region.

**Limitations of existing perturbation-based sensitivity definitions.** Although intuitively appealing, the above perturbation-based sensitivity notions have fundamental limitations when applied to *continuously optimizing adversarial perturbations* in decision-based attacks. While the sensitivity scores in Occlusion, SRA, and PAR are real-valued, they do not provide guidance on how to adjust perturbation strength in a smooth, continuous manner.

For Occlusion and SRA, the sensitivity $s_{w,h}$ indicates the drop in confidence when pixel $(w, h)$ is perturbed, but it does not specify how the perturbation strength at each pixel should be rescaled relative to others to make the global adversarial perturbation more effective, e.g., to reduce the decision-boundary. For PAR, $S_{\text{PAR}}(b)$ gives the maximal compression ratio that preserves attack success for each patch $b$, but this value still cannot serve as a continuous scaling factor for perturbation refinement. Reducing perturbation strength in a patch exactly to its PAR threshold produces an adversarial example that is only barely successful; any further reduction in other regions may cause immediate attack failure. This makes smooth, iterative refinement essentially infeasible.

In summary, these sensitivity definitions characterize *local relationships* between perturbation changes and model outputs (scores or labels), but they do not explicitly capture how local adjustments influence the *global effectiveness* of the perturbation. For decision-based black-box attacks, we argue that a more appropriate notion of perturbation sensitivity should be defined in terms of its impact on the overall adversarial effectiveness, as quantified by the decision-boundary $g(d)$.

**Our decision-boundary-based sensitivity definition.** Given a target DNN model $f$, its indicator function $I(\cdot)$, an original image $x$, and an initial perturbation $d$ that successfully deceives the model (i.e., $I(x + d) = 1$), we define the model's adversarial perturbation sensitivity as a tensor $S \in \mathbb{R}^{C \times W \times H}$, where each element $s_{c,w,h} \geq 0$ represents the sensitivity weight of pixel $(c, w, h)$. In the decision-based setting, we seek a transformed perturbation $S \cdot d$ (element-wise product) that minimizes the decision boundary $g(S \cdot d)$. To ensure that the overall perturbation strength remains unchanged, we enforce an $\ell_2$-norm constraint $\|S \cdot d\|_2 = \|d\|_2$. Accordingly, the task of optimizing perturbations based on sensitivity becomes

$$\arg\min_S \; g(S \cdot d), \quad \text{subject to } \|S \cdot d\|_2 = \|d\|_2. \quad (8)$$

**SeRI perturbation refinement**

Eq. 8 defines a continuous optimization problem over a high-dimensional space $S \in \mathbb{R}^{C \times W \times H}$, which makes the problem especially challenging to solve. To manage this complexity, we adopt an iterative region-splitting strategy. We start from the initial region $b^0 = \{1{:}C, 1{:}W, 1{:}H\}$ and maintain a set of non-overlapping blocks $\mathcal{B}^i$ at iteration $i$, with $\mathcal{B}^0 = \{b^0\}$. At each iteration, we adjust the perturbation only within a single block and then subdivide this block into smaller sub-regions. In practice, each selected block is split into four equal sub-regions; Appendix B.1 shows that the efficiency of SeRI is largely insensitive to this choice.

Given the current perturbation $d^i$ and block set $\mathcal{B}^i$, we select one region to optimize based on its local $\ell_2$-norm: we set $b^* = \arg\max_{b \in \mathcal{B}^i} \|d^i_{[b]}\|_2$, where $d^i_{[b]}$ denotes the restriction of $d^i$ to region $b$. This heuristic prioritizes regions with larger local perturbation, which have greater potential for perturbation reduction. After updating $d^i$ within $b^*$, we replace $b^*$ in $\mathcal{B}^i$ by its four sub-regions to obtain $\mathcal{B}^{i+1}$, enabling progressively finer control.

Once a region $b^*$ is selected, we assess its sensitivity by measuring how local rescaling of the perturbation in $b^*$ changes the overall decision boundary. Let $M_{b^*} \in \{0,1\}^{C \times W \times H}$ be the binary mask of $b^*$, where entries inside $b^*$ are 1 and 0 elsewhere. We construct three candidate perturbations:

$$
\begin{aligned}
d_0^i &= d^i, \\
d_1^i &= \frac{\|d^i\|_2}{\|d^i + (\check{k} - 1)\, d^i \odot M_{b^*}\|_2} \big( d^i + (\check{k} - 1)\, d^i \odot M_{b^*} \big), \\
d_2^i &= \frac{\|d^i\|_2}{\|d^i + (\hat{k} - 1)\, d^i \odot M_{b^*}\|_2} \big( d^i + (\hat{k} - 1)\, d^i \odot M_{b^*} \big),
\end{aligned}
\tag{9}
$$

where $0 < \check{k} < 1 < \hat{k}$ and $\odot$ denotes element-wise multiplication. To keep the overall perturbation strength unchanged, all candidates are normalized to have the same $\ell_2$-norm as $d^i$. We then choose

$$
j^\star = \arg\min_{j \in \{0,1,2\}} g\big(d_j^i\big), \text{ and update } d^{i+1} = d_{j^\star}^i.
\tag{10}
$$

If $j^\star = 1$, the perturbation in $b^*$ is reduced; if $j^\star = 2$, it is enhanced; if $j^\star = 0$, no change is applied, indicating that $b^*$ is already close to locally optimal.

This adaptive update strategy guarantees a monotonic decrease of the decision-boundary distance at each iteration and progressively steers the perturbation of each pixel toward its optimal sensitivity level under the $\ell_2$ constraint. Theoretical justifications of this monotonic improvement and convergence to a stationary perturbation are provided in Appendix A. The complete procedure of SeRI is summarized in Algorithm 1 of Appendix B.

**Comparing decision boundaries of candidate perturbations**

In Eq. 10, SeRI updates the current perturbation by selecting the best candidate among $d_0$, $d_1$, and $d_2$ based on their decision-boundary. This requires a query-efficient procedure to compare the decision boundaries. We adopt the Approximation Decision Boundary Approach (ADBA) Wang et al. (2025), which is specifically designed for low-cost decision-boundary comparisons (see Appendix B.2 for details). As a result, SeRI is highly query-efficient. For example, as shown in Figure 2(a), optimizing a single perturbation over 504 SeRI iterations consumes only 2001 queries in total, i.e., about four queries per iteration ($2001/504 \approx 3.97$).

The effectiveness of ADBA relies on several standard assumptions, SeRI, which builds upon ADBA, also inherits these assumptions (stated in detail in Appendix A.1): (1) local Lipschitz continuity of the decision boundary, (2) locally bounded curvature of the decision boundary, and (3) deterministic hard-label outputs of the target model. Assumptions (1) and (2) are mild and are typically satisfied by modern deep networks such as CNNs and Vision Transformers. Assumption (3), however, rules out randomized defenses (e.g., randomized smoothing or random input transformations Xie et al. (2018); Raff et al. (2019)), and thus SeRI is not intended for such settings. But for other defenses that would not break these three assumptions, such as adversarial training (AT) Zagoruyko (2016) and Lipschitz-based defenses Tsuzuku et al. (2018); Araujo et al. (2023)

## 5 EXPERIMENTS

SeRI improves the perturbations generated by base attackers (e.g., HSJA) by identifying and refining sensitive regions through additional queries. To be effective, the combined "Attacker + SeRI" pipeline must offer better overall query efficiency than the base attacker alone. SeRI is also expected to outperform other sensitivity-aware refinement methods in terms of query usage. To verify these questions, SeRI is evaluated on two famous image classification datasets. SeRI is further compared against multiple decision-based and sensitivity-based attack methods in both targeted and non-targeted settings. All experiments were performed using an Intel Xeon Gold 6330 CPU and four NVIDIA GeForce RTX 4090 GPUs with PyTorch 2.3.0, Torchvision 0.18.0, and Python 3.11.5.

### 5.1 EXPERIMENT SETTINGS

**Competing approaches**. We compare the performance of SeRI with four well-known decision-based attacks, including HSJA, Chen et al. (2020a), CGBA Reza et al. (2023), RayS Chen & Gu

Table 1: Average (median) $\ell_2$ perturbation norms for targeted and non-targeted attacks on ImageNet using a VGG model.

| Total Query | Non-targeted attacks | | | Targeted attacks | | |
|---|---|---|---|---|---|---|
| | 2,000 | 5,000 | 10,000 | 2,000 | 5,000 | 10,000 |
| HSJA | 8.18(5.29) | 4.43(2.51) | 3.39(1.91) | 72.3(66.5) | 50.6(36.9) | 33.7(20.8) |
| HSJA+PAR | 6.65(3.61) | 3.88(2.35) | 3.08(1.82) | 53.0(50.1) | 35.4(32.3) | 20.4(14.0) |
| HSJA+SeRI | 6.47(3.55) | 3.63(2.22) | 2.85(1.51) | **48.3(46.5)** | **32.0(29.9)** | **18.5(12.3)** |
| CGBA | 3.91(2.02) | 1.91(1.10) | 1.19(0.75) | 77.4(74.9) | 58.8(56.4) | 40.2(33.1) |
| CGBA+PAR | 2.81(1.55) | 1.46(0.85) | 1.03(0.64) | 58.3(55.1) | 39.4(37.8) | 23.1(15.6) |
| CGBA+SeRI | 2.92(**1.39**) | **1.36(0.70)** | **0.96(0.54)** | 53.0(50.0) | 36.0(33.2) | 21.1(13.3) |
| RayS | 5.14(3.50) | 3.54(2.27) | 2.72(1.76) | - | - | - |
| RayS+PAR | 3.48(2.38) | 2.23(1.37) | 1.75(1.07) | - | - | - |
| RayS+SeRI | 3.46(2.29) | 2.16(1.27) | 1.56(0.92) | - | - | - |
| ADBA | 4.04(2.77) | 3.03(1.94) | 2.44(1.53) | - | - | - |
| ADBA+PAR | 2.82(1.74) | 2.02(1.19) | 1.67(0.99) | - | - | - |
| ADBA+SeRI | **2.78**(1.71) | 1.95(1.07) | 1.49(0.87) | - | - | - |

Table 2: Average (median) $\ell_2$ perturbation norms for targeted and non-targeted attacks on ImageNet using a ViT model.

| Total Query | Non-targeted attacks | | | Targeted attacks | | |
|---|---|---|---|---|---|---|
| | 2,000 | 5,000 | 10,000 | 2,000 | 5,000 | 10,000 |
| HSJA | 13.2(9.47) | 6.78(4.27) | 4.18(2.88) | 34.3(30.0) | 16.0(15.2) | 8.09(7.94) |
| HSJA+PAR | 9.62(6.19) | 4.98(3.33) | 3.85(2.46) | 27.7(24.1) | 13.0(9.13) | 6.35(5.54) |
| HSJA+SeRI | 9.02(4.40) | 4.13(1.95) | 3.25(2.21) | **26.9(23.1)** | 12.0(8.81) | 6.15(5.27) |
| CGBA | 4.59(3.13) | 2.33(1.52) | 1.59(1.05) | 36.6(31.3) | 14.1(10.3) | 5.80(4.73) |
| CGBA+PAR | 3.67(2.36) | 2.08(1.45) | 1.40(0.94) | 29.3(25.6) | 12.4(8.42) | 4.79(4.26) |
| CGBA+SeRI | **3.53(1.99)** | **1.89(1.29)** | **1.27(0.85)** | 29.0(24.0) | **11.9(8.09)** | **4.61(3.91)** |
| RayS | 10.7(5.24) | 7.20(3.50) | 5.15(2.79) | - | - | - |
| RayS+PAR | 7.05(3.75) | 4.62(2.61) | 3.33(2.02) | - | - | - |
| RayS+SeRI | 7.01(3.69) | 4.52(2.00) | 3.20(1.55) | - | - | - |
| ADBA | 7.56(5.06) | 5.03(3.63) | 3.65(2.71) | - | - | - |
| ADBA+PAR | 5.67(3.65) | 3.87(2.60) | 2.82(1.95) | - | - | - |
| ADBA+SeRI | 5.53(3.62) | 3.59(2.10) | 2.39(1.26) | - | - | - |

(2020), ADBA Wang et al. (2025), and PAR Shi et al. (2022). HSJA and CGBA operate under the $\ell_2$-norm constraint, with CGBA representing the state of the art among such attacks. Additionally, both methods support both targeted and non-targeted attack scenarios. Meanwhile, RayS and ADBA are leading methods with respect to the $\ell_\infty$-norm setting. Building on RayS, ADBA leverages an approximate decision boundary to enable efficient perturbation comparisons, significantly reducing query usage. Due to their $\ell_\infty$-norm constraint, both methods are limit to non-targeted attacks Chen & Gu (2020); Wang et al. (2025). Although RayS and ADBA are originally designed for the $\ell_\infty$-norm attacks, SeRI can still be applied to refine their perturbations, enabling improved performance under the $\ell_2$-norm constraint. Similar to SeRI, PAR is a region-sensitivity-aware perturbation optimizer for the decision-based setting. It can serve as a noise-initialization module for other decision-based attacks, enhancing their noise-compression capability.

We evaluate SeRI across 12 attack configurations: the four base attackers (HSJA, CGBA, RayS, and ADBA), their respective "Attacker + SeRI" variants, and their "Attacker + PAR" counterparts. All configurations are assessed based on attack performance under the $\ell_2$-norm constraint.

**Benchmark datasets and models**. we conduct comprehensive experiments on two datasets using three representative models: ImageNet Deng et al. (2009) with VGG19 Simonyan & Zisserman (2015) and the Vision Transformer (ViT) Dosovitskiy (2020) architecture, and CIFAR-100 Krizhevsky et al. (2009) with an adversarial trained WideResNet model from Wang et al. (2023).

**Hyperparameter settings**. We adopt the recommended hyperparameter settings in Chen et al. (2020a); Chen & Gu (2020); Shi et al. (2022); Reza et al. (2023); Wang et al. (2025). Specifically,

for all four competing algorithms and SeRI, the decision boundary search tolerance $\tau = 10^{-5}$. We also follow hyperparameter settings of PAR in Shi et al. (2022) and set the initial and minimum patch size to 56 and 1, respectively. In SeRI, according to the parameter sensitive analysis in Appendix C, the thresholds $\check{k} = 0.2$ and $\hat{k} = 1.8$, and the query budget percentage $P$ is set to $P = 20\%$.

## 5.2 EXPERIMENT RESULTS

For each model, we randomly select 1000 test images. Tables 1 and 2 present the average and median $\ell_2$-norms achieved by all competing approaches under query budgets of 2,000, 5,000, and 10,000, for both non-targeted and targeted attacks. The best values are highlighted in bold.

**Main results**. As shown in Table 1, CGBA+SeRI achieves the best non-targeted attack performance at both 5,000 and 10,000 queries, outperforming others in terms of average and median $\ell_2$-norms. At 2,000 queries, it achieves the best median performance, while ADBA+SeRI obtains the lowest average $\ell_2$ norm. For targeted attacks, HSJA+SeRI consistently achieves the best performance across all query budgets. As evidenced in Table 2, CGBA+SeRI consistently outperforms other methods in both non-targeted and targeted attacks at 5,000 and 10,000 queries, achieving the lowest average and median $\ell_2$ norms. The only exception is at 2,000 queries for targeted attacks, where HSJA+SeRI performs better.

The results show that the "Attacker + SeRI" variants consistently deliver the best performance across all attack settings. For any given base attacker (e.g., CGBA), combining it with SeRI outperforms both the attacker alone and its "Attacker + PAR" counterpart. Particularly, for the targeted attack results in Table 1, "Attacker + SeRI" achieves superior performance across all query budgets. These results confirm that SeRI's region sensitivity estimation significantly enhances the query efficiency of decision-based attacks.

**Results on adversarially trained WRN model**. Similar to the results on non-adversarially trained VGG and ViT models, Table 3 shows that CGBA+SeRI achieves the best performance at 2,000, 5,000, and 10,000 queries. Additionally, in Table 3, "Attacker + SeRI" reduces the $\ell_2$-norm of perturbations by approximately 30% compared to "Attacker + PAR". It also reduces the $\ell_2$-norm by 15% compared to "Attacker + PAR", as evidenced in Tables 1 and 2. These results indicate that SeRI provides a greater performance gain on adversarially trained WRN models than on non-adversarially trained models.

Table 3: Average (median) $\ell_2$ perturbation norms for non-targeted attacks on CIFAR100 using a WideResNet model.

| Total Query | 2,000 | 5,000 | 10,000 |
|---|---|---|---|
| HSJA | 3.26(2.22) | 1.75(1.15) | 1.26(0.88) |
| HSJA+PAR | 2.59(1.74) | 1.52(0.79) | 1.18(0.83) |
| HSJA+SeRI | 2.08(1.41) | 1.40(0.70) | 1.13(0.63) |
| CGBA | 2.20(1.54) | 1.48(1.01) | 1.19(0.84) |
| CGBA+PAR | 1.71(1.14) | 1.22(0.73) | 1.07(0.65) |
| CGBA+SeRI | **1.48(0.86)** | **1.14(0.67)** | **1.02(0.62)** |
| RayS | 3.17(2.22) | 2.68(1.84) | 2.49(1.69) |
| RayS+PAR | 2.54(1.79) | 2.29(1.70) | 2.15(1.49) |
| RayS+SeRI | 1.79(1.28) | 1.55(1.08) | 1.44(1.01) |
| ADBA | 2.93(2.13) | 2.62(1.77) | 2.46(1.67) |
| ADBA+PAR | 2.41(1.80) | 2.18(1.63) | 2.09(1.50) |
| ADBA+SeRI | 1.74(1.23) | 1.50(1.09) | 1.43(1.04) |

This performance advantage is likely due to the increased difficulty of attacking adversarially trained WRN models, which demands more precise and adaptive perturbation optimization. PAR's binary decision mechanism is less effective against the stronger defenses of adversarially trained WRN models. In contrast, SeRI's fine-grained sensitivity search enables significantly better performance in this challenging setting.

## 5.3 HEATMAPS GENERATED BY SERI

Figure 2 shows the heatmaps over sensitivity regions generated by SeRI across different optimization iterations and queries. More heatmaps generated by SeRI and PAR can be found in Appendix D. In these heatmaps, regions with stronger red intensity indicate areas where SeRI increases perturbation strength, while regions with stronger blue intensity mark non-sensitive areas where perturbation strength is reduced. In Figure 2-(a), the class 'Lacewing' is initially misclassified as 'Walking Stick'. With more iterations, the high-intensity regions in the heatmap increasingly converge and concentrate on the lacewing itself. This example demonstrates the precise thermal grading capability of our SeRI method, which is superior than competing PAR method (see Appendix D for details). At

the 504th iteration, the background is completely marked as blue low-intensity regions, most of the lacewing area is highlighted in intense red, indicating high sensitivity. The transitional areas, including the lacewing's wings and boundaries with the background, are marked in orange, corresponding to moderate-intensity regions. This precise thermal grading not only accelerates perturbation optimization but also aligns with human perception, making it highly interpretable.

Similarly, in Figure 2-(b), the final heat regions converge on the bald eagle's head and left wing areas. In the targeted attack shown in Figure 2-(c), the heatmap concentrates primarily on the Minivan's wheel and door regions in the target image, which enables successful perturbations that misclassify the bald eagle as a Minivan. Note that the interpretive patterns suggested by these heatmaps are secondary effects of the optimization process and should not be interpreted as a complete explanation of model behavior.

### 5.4 FAILURE CASES AND WHEN SERI UNDER-PERFORMS

While SeRI generally improves decision-based attack performance, its effectiveness can vary across different conditions. First, SeRI may under-perform on images with weak or diffuse salient structure, such as cluttered or texture-dominated scenes. In these cases, regional sensitivity becomes less informative, and SeRI provides only modest gains over the base attacker. Second, SeRI requires a moderate query budget to reliably estimate regional importance. Under very low query budgets (e.g., fewer than 50 queries), the region partition becomes too coarse for meaningful refinement, leading to diminishing improvements. Third, similar to all decision-based attacks, SeRI assumes stable outputs from the target model. Therefore, it is not applicable to randomized or stochastic defenses that introduce noise into model predictions, since such randomness breaks the boundary-consistency assumptions required for reliable sensitivity estimation. Recognizing these limitations helps clarify when SeRI is most effective and guides future extensions.

## 6 CONCLUSION AND FUTURE WORK

Sensitive regions in images play a crucial role in determining adversarial vulnerability. However, under strict decision-based black-box settings, it is challenging to identify such regions using only top-1 model predictions. To address this problem, we introduced a novel definition of region sensitivity based on perturbation decision boundaries. This formulation provides a principled and fine-grained way to characterize regional importance and is naturally suited for decision-based attacks. Building on this definition, we proposed SeRI, a new sensitivity-guided decision-based attack framework that adaptively allocates perturbations across regions. Extensive experiments demonstrate that SeRI not only improves the attack performance of state-of-the-art decision-based methods but also produces heatmaps that reliably highlight sensitive image areas, offering clear interpretability benefits.

While our study focuses on image classification, the core idea of SeRI is general and can be extended to other vision tasks such as object detection and semantic segmentation. For multi-output models, SeRI can be adapted by replacing the current decision-boundary–based perturbation comparison with score-based or multi-output perturbation comparisons. This allows SeRI to estimate region sensitivity by examining how perturbations affect task-specific outputs. Exploring these extensions, as well as studying SeRI under stronger or task-specific defenses, represents a promising direction for future research.

### ACKNOWLEDGMENTS

This research is supported by the China Scholarship Council (CSC, Grant No. 202506470044) and partially supported by the BUPT Excellent Ph.D. Students Foundation (Grant No. CX20241003). This research is supported by A*STAR Career Development Fund (Project No. C243512010). The authors thank the anonymous reviewers for their insightful and constructive comments on an earlier version of this manuscript.

## THE USE OF LARGE LANGUAGE MODELS

During the preparation of this manuscript, large language models (LLMs) were used sparingly to enhance grammar, clarity, and readability. All conceptual contributions, technical methods, analyses, and experimental results are entirely original and were developed solely by the authors. The authors have carefully verified the accuracy of all claims and take full responsibility for the content of this paper, in alignment with the ICLR Code of Ethics.

## ETHICS STATEMENT

This research advances the field of AI security by developing efficient query-based black-box adversarial attacks to explore vulnerabilities in deep learning models. As with all security research, there is a potential for dual use, as these techniques could be misused to threaten AI system integrity. However, we firmly believe that proactively uncovering and understanding these vulnerabilities is a critical step towards building more robust and reliable AI systems. By addressing these weaknesses, this work contributes to building more reliable and trustworthy AI, enhancing security across a wide spectrum of applications. Furthermore, we emphasize that SeRI is designed solely for research into adversarial robustness, with the goal of advancing defenses and ensuring the safe deployment of machine learning systems.

## REPRODUCIBILITY STATEMENT

We have made every effort to ensure the reproducibility of our results. The code is provided via the link in the Abstract as well as in the supplementary material. In Section 4, we further provided the full algorithm pseudocode. In Section 5.1, detailed parameter settings and experiment environment are elucidated.

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

# A  THEORETICAL ANALYSIS OF SERI

In this appendix, we provide a formal theoretical analysis of SeRI to analyze its convergence and optimality properties. Let $g(d)$ denote the decision-boundary corresponding to perturbation direction $d$ as defined in eq.4. A smaller value of $g(d)$ implies a more effective adversarial perturbation.

## A.1  ASSUMPTIONS

To analyze the convergence and optimality properties of SeRI, we introduce the following standard assumptions commonly used in decision-based attack theory.

**Assumption 1 (Local Lipschitzness of the decision boundary).**  There exist $L > 0$ such that for all perturbations $d_1, d_2$ in a neighborhood of the search trajectory:

$$|g(d_1) - g(d_2)| \leq L\|d_1 - d_2\|_2.$$

**Assumption 2 (Bounded curvature of the decision boundary).**  Let $\kappa = \|\nabla^2 f(x)\|$ denote the local curvature of the classifier's decision boundary. There exists a constant $C > 0$ such that $\kappa \leq C$ in the neighborhood of interest.

**Assumption 3 (Deterministic hard-label output of the target model).**  The classifier $f$ is query-stable: for any input $z$, repeated queries to the black-box oracle always return the same top-1 label, i.e., $f(z)$ is deterministic.

In particular, there is no randomized defense mechanism such as randomized smoothing, random input transformations (e.g., random translation, rotation, crop), or stochastic ensembling that would change the hard label between queries.

**Lemma 1** (Stable ADBA comparison).  *Under Assumption 3, the outcome of ADBA's comparison between any two perturbations $d_1, d_2$ is deterministic: running ADBA multiple times with the same hyperparameters and query budget always yields the same ordering between $g(d_1)$ and $g(d_2)$.*

*Proof.* ADBA is a deterministic algorithm once the sequence of oracle outputs is fixed. Under Assumption 3, the hard-label oracle is deterministic, so the entire sequence of labels queried during ADBA is fixed. Therefore, the returned decision boundary estimates and the induced ordering between $g(d_1)$ and $g(d_2)$ are also deterministic. □

These assumptions are mild and standard: Assumption 1 holds for ReLU, ViT, and CNN classifiers almost everywhere; Assumption 2 states that the decision boundary has locally bounded curvature; Assumption 3 follows from the concentration guarantees of ADBA Wang et al. (2025).

Assumption 3 acknowledges that SeRI, like all decision-based black-box attacks, may degrade or even fail under randomized defenses such as random cropping, random resizing, randomized smoothing, or stochastic input transformations. This is not a limitation specific to SeRI: in the decision-based setting, the attacker only has access to a single hard label per query, and it is fundamentally impossible to detect or compensate for stochastic transformations applied inside the black-box oracle. Consequently, all existing decision-based attacks, including HSJA, CGBA, RayS, ADBA, and region-based baselines such as PAR, are equally unable to guarantee stable comparisons under such randomized defenses. Our assumption therefore reflects an inherent limitation of the decision-based black-box threat model rather than a deficiency of SeRI itself.

## A.2  SERI CAN GENERATE FINE-GRAINED SENSITIVITY HEATMAPS

Here, we demonstrate that our SeRI approach can produce fine-grained sensitivity heatmaps. Suppose SeRI performs $I$ iterations to generate such a heatmap. In each iteration, it either increases the perturbation strength within a selected region by a factor $k_1 = 1.8$, or decreases it by a factor $k_2 = 0.2$. To analyze this process, we reformulate the problem as follows: Given any initial perturbation pixel $\tilde{p} \in [-1, 1]$ with $\tilde{p} \neq 0$, we aim to show that, after $I$ iterations of multiplicative updates that either $\tilde{p} \leftarrow 1.8 \cdot \tilde{p}$ or $\tilde{p} \leftarrow 0.2 \cdot \tilde{p}$, the perturbation can be adjusted to approach any target

value $\tilde{p}^* \in [-1, 1]$. This guarantees that SeRI can converge to an optimal perturbation configuration consistent with fine-grained sensitivity patterns.

**Perturbation Adjustability of SeRI.** Let $\tilde{p}_0 \in [-1, 1] \setminus \{0\}$ be an initial perturbation value, and let $\tilde{p}^* \in [-1, 1]$ be an arbitrary target value. Under the multiplicative update rule:

$$\tilde{p}_{i+1} = \begin{cases} 1.8 \cdot \tilde{p}_i & \text{(increase)}, \\ 0.2 \cdot \tilde{p}_i & \text{(decrease)}, \end{cases} \tag{11}$$

for any $\epsilon > 0$, there exists a finite sequence of $I$ such updates such that:

$$|\tilde{p}_I - \tilde{p}^*| < \epsilon. \tag{12}$$

**Proof.** We prove the theorem in two parts: (1) by showing that the ratio $\tilde{p}_I / \tilde{p}_0$ can approximate any positive real number arbitrarily closely using multiplicative updates, and (2) by extending this to match the signed target value $\tilde{p}^*$.

Step 1: Logarithmic Density of the Multiplicative Process. After $I$ updates of comprising $m$ increases ($\times 1.8$) and $n = I - m$ decreases ($\times 0.2$), the perturbation becomes:

$$\tilde{p}_I = \tilde{p}_0 \cdot (1.8)^m \cdot (0.2)^n \tag{13}$$

Taking logarithms:

$$\log \tilde{p}_I = \log |\tilde{p}_0| + m \log 1.8 + n \log 0.2 \tag{14}$$

Let $\alpha = \log 1.8$ and $\beta = \log 0.2$. Then:

$$\log \left( \frac{|\tilde{p}_I|}{|\tilde{p}_0|} \right) = m\alpha + n\beta \tag{15}$$

Since $\alpha/\beta \notin \mathbb{Q}$, the set of linear combinations $\{m\alpha + n\beta \mid m, n \in \mathbb{Z}\}$ is dense in $\mathbb{R}$ (this is a standard result from Diophantine approximation Hardy & Wright (1979)).

Exponentiating both sides, the set $\{(1.8)^m (0.2)^n \mid m, n \in \mathbb{Z}\}$ is dense in $\mathbb{R}^+$. Thus, for any positive target ratio $r = |\tilde{p}^*|/|\tilde{p}_0|$ and any $\delta > 0$, there exist integers $m, n$ such that:

$$|(1.8)^m (0.2)^n - r| < \delta \tag{16}$$

Step 2: Matching Sign and Closeness

Choose $\delta = \epsilon/|\tilde{p}_0|$. Then:

$$|\tilde{p}_0 \cdot (1.8)^m (0.2)^n - |\tilde{p}^*|| < \epsilon \tag{17}$$

Now, since all multiplicative updates preserve the sign of $\tilde{p}_0$, we ensure that:

If $\tilde{p}^* > 0$, use $\tilde{p}_0 > 0$ If $\tilde{p}^* < 0$, use $\tilde{p}_0 < 0$

This ensures the **sign of $\tilde{p}_I$ matches that of $\tilde{p}^*$**.

Hence:

$$|\tilde{p}_I - \tilde{p}^*| < \epsilon \tag{18}$$

Since $m$ and $n$ are finite, the total number of operations $I = m + n$ is also finite. The corresponding update sequence can be constructed by applying $m$ increases and $n$ decreases in any order, due to the commutativity of multiplication.

**Conclusion.** Therefore, for any $\epsilon > 0$, any initial $\tilde{p}_0 \in [-1, 1] \setminus \{0\}$, and any target $\tilde{p}^* \in [-1, 1]$, a finite sequence of updates exists such that:

$$|\tilde{p}_I - \tilde{p}^*| < \epsilon \tag{19}$$

### A.3 Monotonic Improvement of SeRI Updates

In each iteration $i$, SeRI constructs three candidate perturbations: $\{d^i, \check{d}^i, \hat{d}^i\}$, where $\check{d}^i$ and $\hat{d}^i$ are obtained by scaling the perturbation on the selected region $b^i$ by factors $\check{k} < 1 < \hat{k}$ (followed by $\ell_2$-norm to match $\|d^i\|_2$), as described in Algorithm 1.

**Theorem 1** (Monotonic Decision-Boundary Descent). *Under Assumptions 1-3, the iterates of SeRI satisfy*

$$g(d^{i+1}) \leq g(d^i)$$

*for all iterations $i$.*

Moreover, suppose that the selected region $b^i$ has nonzero sensitivity, i.e.,

$$\frac{\partial g(d^i)}{\partial d^i[b^i]} \neq 0,$$

and that the scaling factors $\check{k} < 1 < \hat{k}$ are chosen sufficiently close to 1 so that the corresponding perturbation changes remain in a local neighborhood of $d^i$ where a first-order descent direction exists. If, in addition, the ADBA tolerance $\tau$ is chosen sufficiently small and thus can successfully compare the decision boundary of $d^i$, $\check{d}^i$ and $\hat{d}^i$, then the inequality is strict:

$$g(d^{i+1}) < g(d^i).$$

However, if the scaling factors are chosen too close to 1, then the perturbation updates in each refinement iteration become excessively small, which slows down the optimization process and reduces the overall attack efficiency. Therefore, the scaling factors should strike a balance: they must be close enough to 1 to ensure that the updates stay within the local descent region, yet not so close that the algorithm makes negligible progress in each iteration. Based on the empirical results reported in Section C, we adopt $\check{k} = 0.8$ and $\hat{k} = 1.2$ as suitable choices that satisfy both requirements.

*Proof.* By Lemma 1 and Assumption 3, the hard-label oracle is deterministic, hence ADBA produces a deterministic approximation $\hat{g}(d)$ of the true decision boundary $g(d)$ for each candidate perturbation. The stopping criterion of ADBA implies that there exists a tolerance $\tau > 0$ such that $|\hat{g}(d) - g(d)| \leq \tau$ for all candidates considered in SeRI at iteration $t$.

Let $d^i$ be the current perturbation and let $d^{i+1}$ denote the candidate selected by SeRI among $\{d^i, \check{d}^i, \hat{d}^i\}$ using ADBA's estimates. Then we have

$$\hat{g}(d^{i+1}) \leq \hat{g}(d^i),$$

which implies

$$g(d^{i+1}) \leq \hat{g}(d^{i+1}) + \tau \leq \hat{g}(d^i) + \tau \leq g(d^i) + 2\tau.$$

Thus, up to an additive error of order $\tau$, the update is non-increasing.

We now show strict descent under the additional conditions. Let $u^i$ denote the perturbation direction that is nonzero only on the selected region $b^*$, so that the effect of scaling on $b^i$ can be parameterized by a scalar $\alpha$ along $u^i$. Consider the one-dimensional function

$$\varphi_i(\alpha) = g(d^i + \alpha u^i).$$

The assumption $\partial g(d^i)/\partial d^i[b^i] \neq 0$ is equivalent to $\varphi_i'(0) \neq 0$. By continuity of $\varphi_i'$ and the local Lipschitz property in Assumption 1, there exists $\alpha_0 > 0$ and a constant $c > 0$ such that, along the descent direction, we have

$$\varphi_i(\alpha) \leq \varphi_i(0) - c\alpha \quad \text{for all } \alpha \in (0, \alpha_0].$$

The multiplicative scaling on region $b^*$ induces two perturbations $\check{d}^i$ and $\hat{d}^i$ which can be written as

$$\check{d}^i = d^i + \alpha_{\check{k}} u^i, \qquad \hat{d}^i = d^i + \alpha_{\hat{k}} u^i,$$

for some scalars $\alpha_{\check{k}}, \alpha_{\hat{k}}$ determined by $\check{k}, \hat{k}$ and the subsequent normalization. By choosing $\check{k} < 1 < \hat{k}$ sufficiently close to 1, we can ensure that both $|\alpha_{\check{k}}|$ and $|\alpha_{\hat{k}}|$ lie within $(0, \alpha_0]$, i.e., the two scaled candidates stay inside the local neighborhood where the first-order descent behaviour of $\varphi_i$ is valid.

Therefore, at least one of the two scaled candidates, say $d'^i \in \{\check{d}^i, \hat{d}^i\}$, satisfies

$$g(d'^i) = \varphi_i(\alpha') \ \le \ \varphi_i(0) - c|\alpha'| \ \le \ g(d^i) - c',$$

for some constant $c' > 0$ that depends on $c$ and the chosen scaling factors. Choosing $\tau < c'/2$ yields

$$g(d^{i+1}) \ \le \ g(d'^i) + 2\tau \ \le \ g(d^i) - c' + 2\tau \ < \ g(d^i),$$

which gives strict descent.

If no region has nonzero sensitivity, then $\varphi_i'(0) = 0$ for all directions considered, and the candidates coincide up to higher-order terms and numerical tolerance. In that case, the above argument shows that the update is non-increasing but not necessarily strictly decreasing. □

Theorem 1 establishes that SeRI is a **greedy descent method** for minimizing the decision boundary distance. This formally justifies the empirical observation that SeRI consistently strengthens the attack.

### A.4 CONVERGENCE OF THE ITERATIVE PERTURBATION PROCESS

We now show that SeRI converges to a fixed point under the above assumptions.

**Theorem 2** (Convergence to Stationary Point). *Assume Theorem 1 holds and that $g(\cdot)$ is lower bounded. Then the sequence $\{g(d^i)\}$ converges, and any limit point $d^\star$ satisfies:*

$$\frac{\partial g(d^\star)}{\partial d[b]} = 0 \quad \text{for all subregions } b.$$

*Proof.* Since $g(d^{i+1}) \le g(d^i)$ and $g$ is non-negative, the sequence is monotonically decreasing and bounded below. Thus it converges. If a limit point $d^\star$ had nonzero partial derivative in some region $b$, the multiplicative update with sufficiently small scaling factor would strictly decrease $g$, contradicting convergence. Hence all directional derivatives vanish. □

This establishes that SeRI converges to a **stationary perturbation** under its multiplicative refinement mechanism.

### A.5 CONSISTENCY OF THE REGION-SENSITIVITY ESTIMATION

Recall that SeRI recursively partitions current region $b$ into four equal sub-regions (Algorithm 2), producing a quad-tree hierarchy. Let $S^*$ denote the true optimal pixel-wise sensitivity map, and let $S_k$ denote the piecewise-constant approximation obtained after $k$ refinement iterations.

To formalize the convergence analysis, we first define the notion of the *cell size* of a region.

**Definition (Cell size).** For any region $b \subset \mathbb{R}^2$, we define its cell size $h(b)$ as the side length of the smallest axis-aligned square containing $b$. Equivalently, $h(b)$ is proportional to the diameter of $b$ up to a constant factor:

$$\text{diam}(b) \le \sqrt{2}\, h(b).$$

Under the $2 \times 2$ recursive split used by SeRI, every refinement iteration reduces the cell size by half. If $h_0$ denotes the initial cell size (corresponding to the entire image domain), then the cell size after $k$ refinement levels is

$$h_k = 2^{-k} h_0.$$

**Regularity assumption.** We assume that $S^*$ is piecewise Hölder continuous of order $\alpha > 0$, meaning that the image domain can be partitioned into finitely many subdomains $\{\Omega_i\}$ such that for each $\Omega_i$,

$$|S^*(x) - S^*(y)| \leq C\|x - y\|^{\alpha}, \qquad \forall x, y \in \Omega_i,$$

for some constant $C > 0$. This assumption allows $S^*$ to have discontinuities across object boundaries or semantic edges, while maintaining smoothness within each piece.

**Theorem 3** (Consistency of Region Refinement). *Let $S_k$ be the region-wise sensitivity estimate produced after $k$ levels of quad-tree refinement. If $S^*$ is piecewise Hölder continuous of order $\alpha > 0$, then*

$$\|S_k - S^*\|_2 = O(2^{-k\alpha}).$$

*Sketch.* Consider any region $b$ at refinement level $k$. Since $S_k$ assigns a constant value to $b$, the approximation error on this region satisfies

$$\sup_{x \in b} |S_k(x) - S^*(x)| \leq C\,h(b)^{\alpha} = C\,h_k^{\alpha},$$

by the Hölder condition on each smooth piece. Summing over the quad-tree cells and noting that the number of cells grows only polynomially while the per-cell error decays as $h_k^{\alpha}$, we obtain

$$\|S_k - S^*\|_2^2 = O(h_k^{2\alpha}) = O(2^{-2k\alpha}),$$

which proves the stated $O(2^{-k\alpha})$ convergence rate in the $L^2$ norm. $\square$

This theorem shows that SeRI's region-refinement strategy is **provably consistent**: as the quad-tree becomes finer, the region-wise sensitivity estimate $S_k$ converges to the true underlying sensitivity $S^*$. In Appendix B.1, we further conduct an empirical comparison among several partitioning strategies ($1 \times 2$, $2 \times 2$, $3 \times 3$, and $4 \times 4$). The results show that the $2 \times 2$ split provides the best trade-off between refinement granularity and query efficiency, while the other strategies produce comparable but slightly inferior results. This empirical evidence supports the theoretical analysis by demonstrating that recursive region refinement is robust to the exact splitting scheme, and that the $2 \times 2$ strategy achieves both strong practical performance and favorable theoretical properties.

## A.6 ROBUSTNESS UNDER MODEL VARIATIONS

Using Assumption 2, we show that ADBA remains robust even under boundary curvature.

**Proposition 1** (ADBA Robustness Under Bounded Curvature). *Let $d_1, d_2$ be two candidate perturbations. Under Assumption 2,*

$$|g(d_1) - g(d_2)| \leq O(\|d_1 - d_2\|_2) + O(\kappa\|d_1 - d_2\|_2^2).$$

*Thus, for perturbations differing only on a single region scaling (as in SeRI), the second-order term is negligible, and ADBA's ranking of candidates is stable.*

*Sketch.* A second-order Taylor expansion of the boundary distance along the perturbation direction yields the claim. $\square$

This establishes that SeRI is robust to model architectural variations, since curvature only affects second-order terms.

## A.7 CONNECTION BETWEEN DECISION-BOUNDARY AND ATTACKABILITY

In this subsection we clarify why minimizing the decision-boundary $g(d)$ is closely aligned with improving the attackability of a model. Recall that for a fixed clean image $x$ and perturbation direction $d$, the decision-boundary distance is defined as

$$g(d) = \min\{r > 0 : I(x + r\,d/\|d\|_2) = 1\},$$

that is, the smallest radius at which the perturbed point leaves the original class (non–targeted attack) or enters the target class (targeted attack).

**Attackability under an $\ell_2$ norm budget.** Suppose an attacker is allowed to use any perturbation with $\ell_2$ norm at most $\epsilon > 0$. For a fixed direction $d$, the attack succeeds within this budget if and only if there exists $r \leq \epsilon$ such that $I(x + r\, d/\|d\|_2) = 1$, which is equivalent to $g(d) \leq \epsilon$. Hence, for a fixed $\epsilon$ we have

$$\mathbf{1}\{\text{attack succeeds within norm } \epsilon\} = \mathbf{1}\{g(d) \leq \epsilon\}.$$

Consequently, among two perturbation directions $d_1$ and $d_2$, if $g(d_1) < g(d_2)$ then for any fixed norm budget $\epsilon$,

$$\mathbf{1}\{g(d_1) \leq \epsilon\} \geq \mathbf{1}\{g(d_2) \leq \epsilon\}.$$

This shows that a smaller decision-boundary is strictly more favorable for achieving successful attacks under the same $\ell_2$ constraint: any budget for which $d_2$ can succeed, $d_1$ can also succeed, while the converse is not true.

**Attackability under random perturbation magnitudes.** The above argument can be generalized to randomized attack procedures. Let $R$ be a non–negative random variable describing the perturbation magnitude produced by some attack algorithm along direction $d$ (for example, the final radius found by a boundary search under a fixed query budget). The attack succeeds if $R \geq g(d)$. Denoting the cumulative distribution function of $R$ by $F_R(r) = \Pr(R \leq r)$, the success probability for direction $d$ is

$$\Pr(\text{success} \mid d) = \Pr(R \geq g(d)) = 1 - F_R\big(g(d)\big).$$

If $F_R$ is strictly increasing, then for any two directions $d_1$ and $d_2$

$$g(d_1) < g(d_2) \quad \implies \quad 1 - F_R\big(g(d_1)\big) > 1 - F_R\big(g(d_2)\big),$$

which means that a smaller decision-boundary distance implies a larger success probability under the same attack procedure and query budget. Therefore, minimizing $g(d)$ is equivalent to maximizing the attack success probability for any fixed distribution of perturbation magnitudes.

**Empirical correlation between $g(d)$ and attackability.** To complement this theoretical argument, we conduct an additional experiment where, for each test image, we record the estimated decision-boundary distance $\hat{g}(d)$ produced by ADBA and the corresponding attack outcome (success or failure) under a fixed norm threshold $\epsilon$. We then group samples by quantiles of $\hat{g}(d)$ and report the empirical attack success rate in each bin. The results, shown in Figure A.7, display a clear monotone trend: bins with smaller $\hat{g}(d)$ exhibit substantially higher attack success rates, while bins with larger $\hat{g}(d)$ have markedly lower success rates. This empirical evidence supports our theoretical analysis, and confirms that SeRI's objective of reducing the approximate decision-boundary distance directly translates into higher attackability under practical query budgets.

# B  PSEUDOCODE OF SERI

In Algorithm 1, SeRI iteratively enhances or reduces the perturbation strength within the selected sub-region to identify whether the selected sub-region is sensitive to perturbation, and then splits the sub-region into four smaller parts. Splitting each region into four parts offers the best trade-off between precision and efficiency. As shown in both our theoretical analysis and the experimental results in Appendix B.1, this strategy achieves superior performance compared to alternative partitioning schemes. The initial region $b_0$ has a shape of $C \times W \times H$. Algorithm 2 splits $b_0$ into four smaller sub-regions, forming the initial sub-region set $\mathcal{B}$ (see Appendix B.1 for details).

In line 2 of Algorithm 1, at each iteration, the sub-region $b^*$ with the hightest $\ell_2$-norm perturbation is selected. Then in lines 3-5, SeRI generates two new perturbations, $\check{d}^i$ and $\hat{d}^i$, by respectively reducing and enhancing the perturbation within sub-region $b^*$. The reduction and amplification multipliers are denoted by $\check{k}$ and $\hat{k}$, respectively. In line 5, the perturbations $\check{d}^i$ and $\hat{d}^i$ are normalized to match the overall perturbation strength of $d^i$, ensuring a fair comparison among $d^i$, $\check{d}^i$ and $\hat{d}^i$ as defined in Eq. 8. In line 6, we use ADBA Wang et al. (2025) to compare the decision boundaries of $d^i$ and $\check{d}^i$ to determine the best perturbation (see Algorithm 3 in Appendix B for details). Specifically, Algorithm 3 takes as input the current best perturbation $d^i$, along with its approximate decision boundaries $R_{\min}$ and $R_{\max}$ (lower and upper bounds of the true decision boundary), as well as a candidate perturbation $\check{d}^i$. The algorithm then outputs the superior perturbation $d_s$ together

---

**Algorithm 1** Perturbation Optimization via Gradient-Free Sensitivity Region Identification

---

**Input**: Original image $x$, initial adversarial example $\tilde{x}^0$, indicator $I(\cdot)$, query budget $Q$;
**Output**: Adversarial sensitive region heatmap $S$, optimized adversarial example $\tilde{x}$;
**Initialization**: Iteration number $i \leftarrow 1$, initial perturbation $d^1 \leftarrow \tilde{x}^0 - x$, approximation decision boundary $R_{\min} \leftarrow 0$, and $R_{\max} \leftarrow 1$; initial region $b_0 \leftarrow \{1 : C, 1 : W, 1 : H\}$, sub-region set $\mathcal{B} \leftarrow$ Algorithm 2 $(b_0)$; perturbation multipliers $\check{k} \leftarrow 0.2$, and $\hat{k} \leftarrow 1.8$;

1:  **while** remaining query budget $> 0$ **do**
2:      $b^* \leftarrow \arg\max\limits_{b \in \mathcal{B}} \|d^i_{[b]}\|_2$
3:      $\check{d}^i, \hat{d}^i \leftarrow d^i.copy(), d^i.copy()$
4:      $\check{d}^i_{[b]}, \hat{d}^i_{[b]} \leftarrow \check{k} \cdot d^i_{[b]}, \hat{k} \cdot d^i_{[b]}$
5:      $\check{d}^i, \hat{d}^i \leftarrow \frac{\|d^i\|_2}{\|\check{d}^i\|_2} \cdot \check{d}^i, \frac{\|d^i\|_2}{\|\hat{d}^i\|_2} \cdot \hat{d}^i$
6:      $d_s, R_{\min}, R_{\max} \leftarrow$ Algorithm 3 $(d^i, R_{\min}, R_{\max}, \check{d}^i)$
7:      **if** $d_s = \check{d}^i$ **then**
8:          $d^{i+1} \leftarrow \check{d}^i$
9:      **else**
10:         $d_s, R_{\min}, R_{\max} \leftarrow$ Algorithm 3 $(d^i, R_{\min}, R_{\max}, \hat{d}^i)$
11:         **if** $d_s = \hat{d}^i$ **then**
12:             $d^{i+1} \leftarrow \hat{d}^i$
13:         **else**
14:             $d^{i+1} \leftarrow d^i$
15:         **end if**
16:     **end if**
17:     $\mathcal{B} \leftarrow (\mathcal{B} \setminus \{b^*\}) \cup$ Algorithm 2 $(b^*)$;
18:     $i \leftarrow i + 1$
19:     $S \leftarrow d^i \oslash d^1$ // $s_{c,w,h} = d^i_{c,w,h}/d^1_{c,w,h}$
20:     $\tilde{x} \leftarrow x + R_{\max} \cdot d^i$
21: **end while**
22: **return** $S, \tilde{x}$

---

**Algorithm 2** Split Perturbation Into Four sub-regions

---

**Input**: Initial region $b_0 \leftarrow \{1 : C, x_1 : x_2, y_1 : y_2\}$;
**Output**: sub-region set of $b_0$;

1:  $x_{\mathrm{mid}} \leftarrow \left\lfloor \frac{x_1 + x_2}{2} \right\rfloor, y_{\mathrm{mid}} \leftarrow \left\lfloor \frac{y_1 + y_2}{2} \right\rfloor$
2:  $b_1 \leftarrow \{1 : C, x_1 : x_{\mathrm{mid}}, y_1 : y_{\mathrm{mid}}\}$
3:  $b_2 \leftarrow \{1 : C, x_{\mathrm{mid}} + 1 : x_2, y_1 : y_{\mathrm{mid}}\}$
4:  $b_3 \leftarrow \{1 : C, x_1 : x_{\mathrm{mid}}, y_{\mathrm{mid}} + 1 : y_2\}$
5:  $b_4 \leftarrow \{1 : C, x_{\mathrm{mid}} + 1 : x_2, y_{\mathrm{mid}} + 1 : y_2\}$
6:  **return** $\{b_1, b_2, b_3, b_4\}$

---

with updated boundary estimates $R_{\min}$ and $R_{\max}$. If $\check{d}^i$ outperforms $d^i$, as determined in line 7-8, this indicates that perturbations within region $b^*$ do not contribute positively to the attack's success. Hence $\check{d}^i$ is deemed the best perturbation among $d^i$, $\check{d}^i$ and $\hat{d}^i$, and the updated perturbation $d^{i+1}$ is set to $\check{d}^i$. In line 10, if $d^i$ performs better than $\check{d}^i$, $d^i$ is further compared with $\hat{d}^i$ using Algorithm 3. In lines 11–15, $d^{i+1}$ is updated to match the better perturbation between $d^i$ and $\hat{d}^i$, based on their decision boundaries. In line 17, the selected region $b^*$ is removed from the sub-region set $\mathcal{B}$ and replaced with its four child sub-regions generated by Algorithm 2. In line 19, the heatmap $S$ is updated by performing element-wise division between $d^i$ and $d^1$. Subsequently in line 20, the current adversarial example $\tilde{x}$ is updated. In lines 21–22, if the query budget is reached, the algorithm stops and returns the sensitive region heatmap $S$ together with the optimized adversarial example $\tilde{x}$ as the final output.

Table 4: Average (median) $\ell_2$ norms for non-targeted attacks on ImageNet using a VGG model.

| Total Query | 2,000 | 5,000 | 10,000 |
|---|---|---|---|
| 1×2 split | 3.292(1.429) | 1.630(0.811) | 1.024(0.569) |
| 2×2 split | **3.234(1.385)** | **1.589(0.771)** | **1.020(0.562)** |
| 3×3 split | 3.290(1.410) | 1.598(0.776) | 1.029(0.569) |
| 4×4 split | 3.311(1.444) | 1.645(0.800) | 1.040(0.570) |

### B.1 SPLIT SELECTED REGION INTO FOUR SUB-REGIONS

To capture adversarial perturbation sensitivity at a fine granularity, the current region $b_0$ is divided into four sub-regions after each iteration. The detailed procedure is shown in Algorithm 2. In line 1, $x_{\mathrm{mid}}$ and $y_{\mathrm{mid}}$ denote the horizontal and vertical midpoints of the current region $b_0$. Lines 2–5 generate four sub-regions $b_1, b_2, b_3, b_4$ based on these midpoints. Line 6 returns the set of newly created sub-regions.

To determine the optimal number of sub-regions per split, we conducted an experimental comparison of a 2×2 split against three alternatives: a 1×2 binary split (direction chosen by aspect ratio), a 3×3 split, and a 4×4 split. The evaluation was performed on 100 images from the ImageNet dataset using a VGG model. We report the average (median) $\ell_2$ perturbation norms achieved by the "CGBA+SeRI" attack. The results, presented in Table 4, demonstrate that the 2×2 split yields the best performance across 2,000, 5,000, and 10,000 total queries. However, the performance of other split settings is comparable, as their norms are only marginally higher (by approximately 3%). This indicates that the overall performance of SeRI is not highly sensitive to the exact number of splits.

### B.2 COMPARE TWO PERTURBATIONS USING APPROXIMATION DECISION BOUNDARY

To compare perturbations effectively in decision-based attack setting, we follow the cutting-edge Approximate Decision Boundary Approach (ADBA) recently proposed in Wang et al. (2025). ADBA avoids the need to precisely compute decision boundaries with high query cost. Instead, it compares two perturbations by identifying a perturbation strength where one successfully fools the model but the other fails. This indicates that the successful perturbation has a smaller decision boundary and is more effective.

The main idea of ADBA is that, it is unnecessary to precisely identify the decision boundaries of two perturbations to compare them. Instead, if we can identify an Approximate Decision Boundary (ADB), such that at this perturbation strength, one perturbation $d_1$ successfully fools the model while another direction $d_2$ fails, then we can infer that the decision boundary of $d_1$, $g(d_1)$, is smaller than that of $d_2$, i.e., $g(d_1) \leq \mathrm{ADB} < g(d_2)$). This implies that $d_1$ outperform $d_2$.

The procedure is summarized in Algorithm 3. In this algorithm, $R_{\max}$ and $R_{\min}$ represent the upper and lower bounds of the ADB for $d_1$. Specifically, $d_1$ successfully fools the model at perturbation strength $R_{\max}$ (i.e., $I(x + R_{\max} \cdot d_1) = 1$, indicating $g(d_1) \leq R_{\max}$), but fails to fool the model at perturbation strength $R_{\min}$ (i.e., $I(x + R_{\min} \cdot d_1) = 0$, indicating $R_{\min} < g(d_1)$). In lines 1-3, if perturbation $d_2$ fails to fool the model at ADB $R_{\max}$, it indicates that its true decision boundary is greater than $R_{\max}$, i.e., $g(d_1) \leq R_{\max} < g(d_2)$. In this case, $d_2$ is less effective than $d_1$ and Algorithm 3 return $d_1$ as the superior perturbation. Otherwise, in lines 4-6, if perturbation $d_2$ successfully fools the model at ADB $R_{\min}$, it indicates that its true decision boundary is smaller than $R_{\min}$, i.e., $g(d_2) \leq R_{\min} < g(d_1)$. This implies that $d_2$ is more effective than $d_1$, and Algorithm 3 returns $d_2$ as the superior perturbation. In lines 7-13, if both $d_1$ and $d_2$ either succeed or fail at the current approximation $R$, then $R_{\max}$ or $R_{\min}$ are updated accordingly to narrow the search interval for $R$. In lines 14–17, if the current value of $R$ leads to a successful attack for one perturbation but not the other, the successful perturbation is returned along with the updated ADBs, $R_{\min}$ and $R_{\max}$. These updated bounds are carried into the next iteration to narrow the search range and reduce the number of queries required in lines 1–6 of Algorithm 3.

Finally, in line 20, if the search interval $R_{\max} - R_{\min}$ becomes smaller than a search tolerance threshold $\tau$, indicating that $g(d_1)$ and $g(d_2)$ are nearly equivalent, it becomes unnecessary to distinguish between them. Hence, the algorithm directly returns $d_2$ along with the current $R_{\min}$ and $R_{\max}$. Returning $d_1$ produces similar results in our experiments.

---

**Algorithm 3** Compare Two Perturbations Using Approximation Decision Boudnary

---

**Input**: Current best perturbation $d_1$ with approximation decision boundaries $R_{\min}, R_{\max}$; candidate perturbation $d_2$, and search tolerance $\tau = 10^{-5}$;
**Output**: Superior perturbation $d_s$ with updated approximation decision boundary $R_{\min}$, $R_{\max}$.

  1: **if** $I(x + R_{\max} \cdot d_2) = 0$ **then**
  2:    **return** $d_1, R_{\min}, R_{\max}$
  3: **end if**
  4: **if** $I(x + R_{\min} \cdot d_2) = 1$ **then**
  5:    **return** $d_2, 0, R_{\min}$
  6: **end if**
  7: **while** $R_{\max} - R_{\min} > \tau$ **do**
  8:    **if** $I(x + R \cdot d_2) = 1$ and $I(x + R \cdot d_1) = 1$ **then**
  9:      $R \leftarrow (R_{\min} + R_{\max})/2$
10:      $R_{\max} \leftarrow R$
11:    **else if** $I(x + R \cdot d_2) = 0$ and $I(x + R \cdot d_1) = 0$ **then**
12:      $R \leftarrow (R_{\min} + R_{\max})/2$
13:      $R_{\min} \leftarrow R$
14:    **else if** $I(x + R \cdot d_2) = 1$ and $I(x + R \cdot d_1) = 0$ **then**
15:      $R_{\max} \leftarrow R$
16:      **return** $d_2, R_{\min}, R_{\max}$
17:    **else if** $I(x + R \cdot d_2) = 0$ and $I(x + R \cdot d_1) = 1$ **then**
18:      $R_{\max} \leftarrow R$
19:      **return** $d_1, R_{\min}, R_{\max}$
20:    **end if**
21: **end while**
22: **return** $d_2, R_{\min}, R_{\max}$

---

## C   PARAMETER SENSITIVITY ANALYSIS

Our SeRI approach summarized in Algorithm 1 introduces several parameters, including the thresholds $\check{k} = 0.2$ and $\hat{k} = 1.8$, as well as the SeRI query budget percentage $P = 20\%$. These parameter settings are based on preliminary experiments with the ImageNet-VGG19 model. Similar results have been observed across other datasets and models.

Table 5: $\ell_2$-norm of perturbation for varying SeRI hyperparameters $(\hat{k}, \check{k})$ under 5,000 query budget.

|  | $\hat{k} = 1.7$ | $\hat{k} = 1.8$ | $\hat{k} = 1.9$ | $\hat{k} = 2.0$ |
|---|---|---|---|---|
| $\check{k} = 0.05$ | 1.261 | 1.258 | 1.257 | 1.262 |
| $\check{k} = 0.10$ | 1.258 | 1.251 | 1.251 | 1.259 |
| $\check{k} = 0.15$ | 1.251 | 1.247 | 1.260 | 1.249 |
| $\check{k} = 0.20$ | 1.243 | **1.239** | 1.246 | 1.255 |
| $\check{k} = 0.25$ | 1.258 | 1.256 | 1.251 | 1.265 |

Table 6: $\ell_2$-norm of perturbation for varying SeRI hyperparameter $P$ under different query budget.

| Total Query$\rightarrow$ | 2,000 | 5,000 | 10,000 | 20,000 |
|---|---|---|---|---|
| $P = 10\%$ | 3.475 | 1.330 | 0.857 | 0.700 |
| $P = 20\%$ | **3.112** | **1.259** | **0.823** | **0.670** |
| $P = 30\%$ | 3.304 | 1.480 | 0.873 | 0.747 |
| $P = 40\%$ | 3.332 | 1.719 | 0.994 | 0.819 |
| $P = 100\%$ | 4.174 | 2.826 | 2.237 | 1.914 |

To evaluate the impact of different parameter settings, we conduct a sensitivity analysis. Specifically, we vary the threshold $\check{k}$ and $\hat{k}$ (Table 5) and the percentage $P$ (Table 6) across a range of values. For

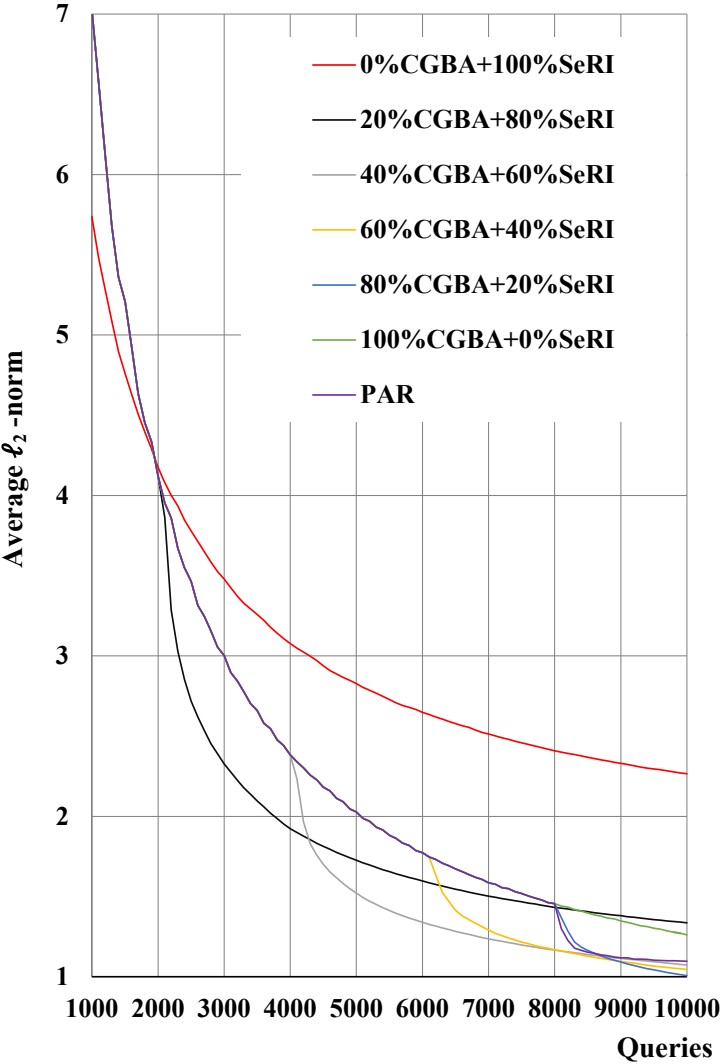

Figure 3: Decrease of the perturbation $\ell_2$-norm under different $P$ settings of SeRI on the ImageNet-VGG model.

each setting, attack performance is assessed using the $\ell_2$-norm of the final perturbation. The total query budget is set to 5,000 for threshold $\check{k}$ and $\hat{k}$. Regarding $P$, we evaluate the attack under query budgets of 2,000, 5,000, 10,000, and 20,000. This analysis examines whether the optimal choice of $P$ depends on the total query budget. A lower $\ell_2$-norm indicates a more effective attack.

We conduct experiments on the ImageNet dataset using VGG19 model for non-targeted attack. This model have been introduced in Subsection 4.1. The results are presented in Tables 5 and 6, with the best values highlighted in bold.

Our experimental results demonstrate that the parameter combination ($\hat{k} = 0.2, \check{k} = 1.8$) and $P = 20\%$ consistently yields the best performance. As shown in Table 5, the setting ($\hat{k} = 0.2, \check{k} = 1.8$) achieves the lowest $\ell_2$-perturbation of 1.239. Additionally, Table 6 confirms that $P = 20\%$ results in the lowest $\ell_2$-perturbation across all four query budget settings.

We also plot the decrease of the perturbation $\ell_2$-norm under different $P$ settings of SeRI on the ImageNet-VGG model (Fig.3). From the figure, we observe that when the query budget is small (before approximately 1900 queries), the setting $P = 0\%$ achieves the lowest $\ell_2$-norm. However, as the query budget increases, the performance of $P = 0\%$ becomes significantly worse. This is expected because $P = 0\%$ corresponds to initializing SeRI with a random perturbation, which allows for a rapid early reduction of the $\ell_2$-norm but provides no meaningful optimization direction for later refinement.

For $P = 20\%, 40\%, 60\%$, and $80\%$, we observe a noticeable "jump" in the $\ell_2$-norm when SeRI begins operating, indicating that SeRI can effectively refine and further optimize the perturbation produced by the base attacker.

The setting $P = 100\%$ corresponds to using the base attacker alone (CGBA) with no SeRI refinement. In this case, no second-stage improvement occurs, and thus the $\ell_2$-norm does not decrease significantly at 10,000 queries.

# D    COMPARISON OF HEATMAPS GENERATED BY SeRI, PAR, LIME, AND SHAP

Figure4 provides a qualitative comparison of seven input images alongside the corresponding heatmaps and perturbation produced by SeRI and PAR. In this experiment, we adopt CGBA as the base attacker and set the total query budget to 10,000. As illustrated in the figure, the proposed SeRI method yields significantly more informative and concentrated heatmaps than PAR, leading to both improved interpretability and attack quality. The advantages of SeRI can be summarized in two major aspects:

1. Semantically concentrated perturbations. SeRI naturally guides perturbations toward semantically meaningful object regions (e.g., a dog's head or a whale's tail), while PAR often generates artifacts scattered across background areas. These background perturbations are visually disruptive and lack semantic relevance. By effectively suppressing such noise, SeRI achieves a substantially reduced perturbation strength and produces adversarial examples that are less perceptible to human observers.

2. Continuous and fine-grained saliency modeling. SeRI provides continuous-valued regional importance estimates: highly influential areas are highlighted in dark red, moderately relevant areas appear in orange or yellow, and unimportant background regions are represented in blue. This continuous sensitivity landscape more faithfully reflects the underlying structure of the model's decision surface. By contrast, PAR applies a binary patch-retention mechanism, either preserving or removing an entire patch, thus failing to capture nuanced differences in regional contributions and often hindering the optimization performance.

Furthermore, the heatmaps generated by SeRI enhance the interpretability of the attack, exhibiting strong consistency with those produced by classical explainable AI (XAI) methods. Figure5 compares the heatmaps generated by SeRI, PAR, LIME, and Kernel SHAP.

LIME is a model-agnostic local explanation method based on superpixel perturbation and sparse regression. In our implementation, LIME is configured with 1,000 perturbation samples and uses superpixel boundaries (yellow contours) to highlight high-importance regions.

Kernel-based SHAP is the black-box variant of SHAP, requiring only model output scores rather than gradients. It estimates Shapley values by solving a locally weighted linear regression. In our experiments, Kernel SHAP is configured with 50 superpixels (SLIC segmentation), 300 sampled coalition evaluations. Regions important to the model are visualized in red, moderately relevant areas in yellow/green, and unimportant areas in blue.

To quantify the agreement between our sensitivity maps and classical XAI methods, we compute the Pearson correlation coefficient (PCC) between the heatmaps produced by SeRI/PAR and those generated by LIME and SHAP. As shown in Table7, SeRI achieves PCC scores of 0.723 with LIME and 0.786 with SHAP values commonly interpreted as indicating strong correlation. In comparison, PAR exhibits noticeably lower correlations (0.610 with LIME and 0.637 with SHAP). These results suggest that SeRI captures more consistent and meaningful saliency structure than PAR, and aligns

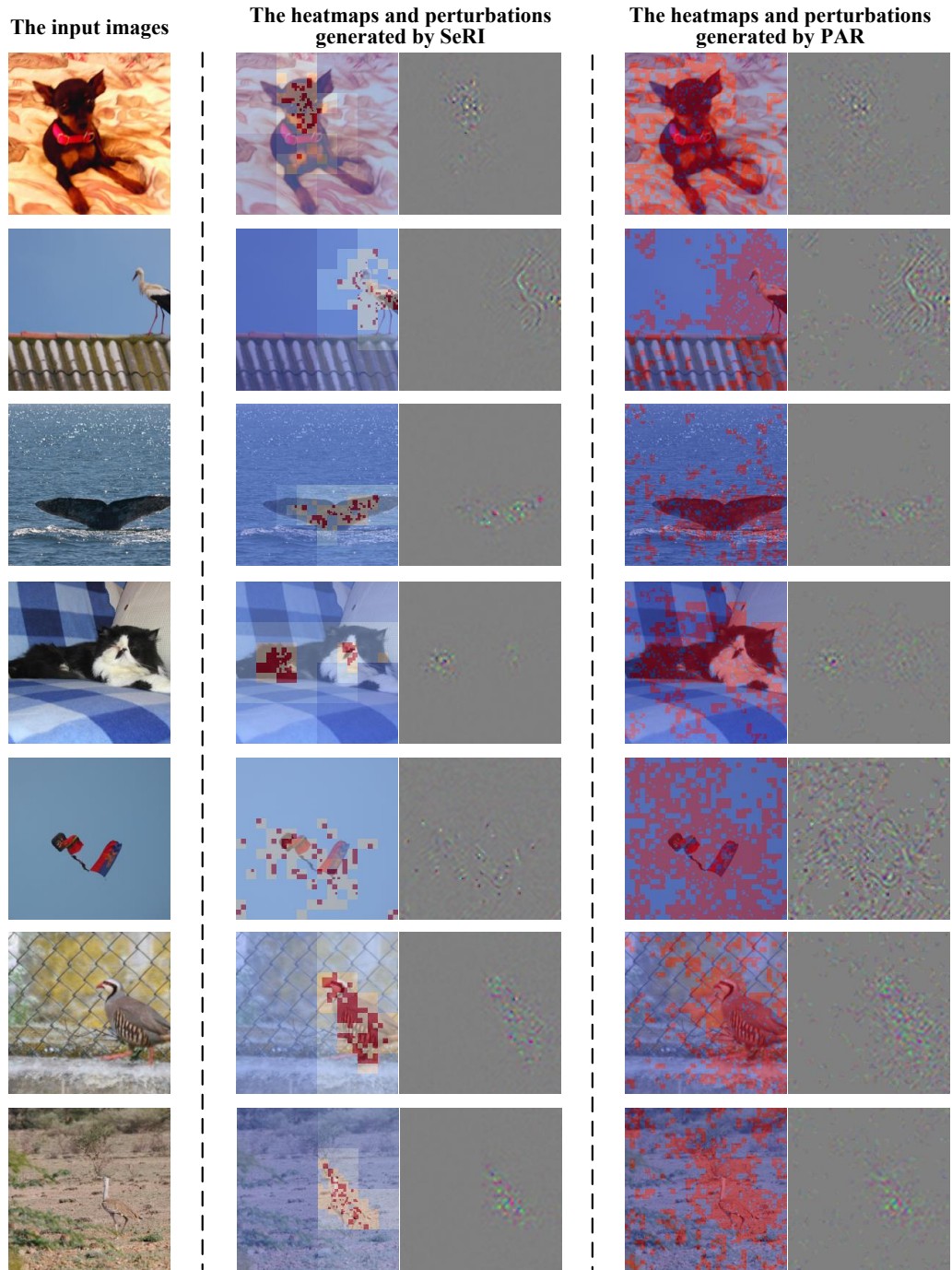

Figure 4: Heatmaps and perturbations generated by SeRI and PAR.

more closely with the explanations provided by established XAI methods, despite relying only on hard-label queries.

This quantitative evidence is further supported by the qualitative comparisons in Figure5. For the first-row dog example, both LIME and SHAP assign the highest importance to the dog's head, and SeRI similarly concentrates its heatmap on this region. A similar pattern is observed in the

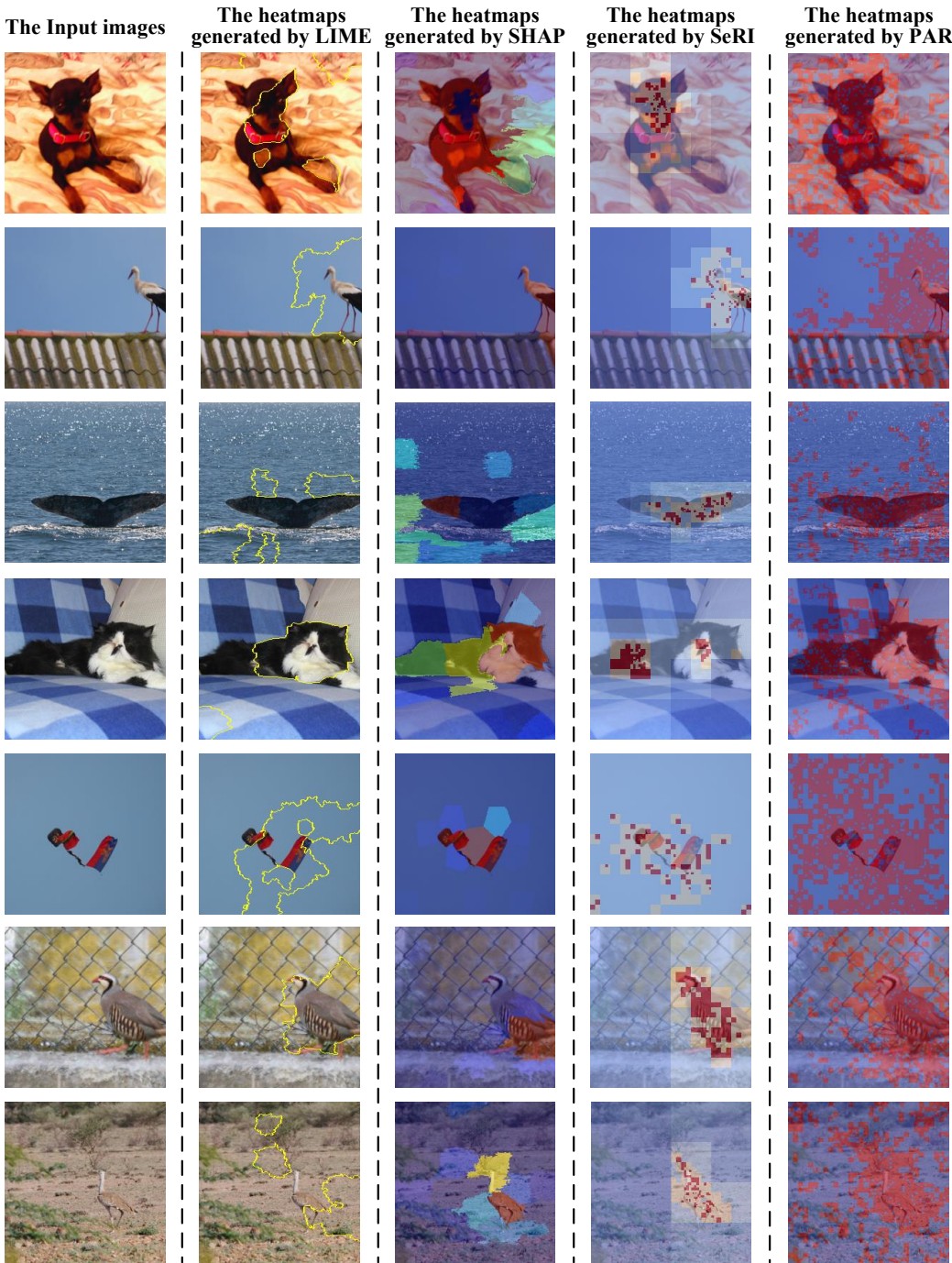

Figure 5: Heatmaps generated by LIME, SHAP, SeRI, and PAR.

second-row bird image: LIME and SHAP highlight the bird's torso, wings, and legs, and SeRI again identifies these same regions with high saliency.

Interestingly, the third-row whale-tail example reveals an even clearer distinction. While LIME and SHAP partially capture the whale's tail but fail to fully localize it, SeRI successfully places nearly all of its saliency on the tail, closely matching the true decision-critical region. This high-

Table 7: Average Pearson Correlation Coefficient (PCC) between the heatmaps produced by SeRI/PAR and those from LIME and SHAP.

| Method | PCC w/ LIME | PCC w/ SHAP |
|--------|-------------|-------------|
| **SeRI** | **0.723** | **0.786** |
| PAR | 0.610 | 0.637 |

lights SeRI's ability to more accurately approximate the model-dependent sensitivity landscape than existing black-box XAI baselines.

Overall, both the quantitative PCC analysis and qualitative heatmap comparisons indicate that SeRI not only delivers more semantically accurate and interpretable region sensitivity estimates than PAR, but also maintains a strong alignment with classical XAI methods such as LIME and SHAP. These results further validate the reliability and effectiveness of SeRI's boundary-driven sensitivity estimation framework.

# E COMPARE ATTACK SUCCESS RATE (ASR), STRUCTURAL SIMILARITY INDEX (SSIM), AND STANDARD DIVISION OF $\ell_2$-NORM ($\ell_2$-STD)

To further verify that SeRI provides advantages beyond $\ell_2$-norm reduction, we conduct additional experiments on three complementary metrics: (1) Attack Success Rate (ASR), (2) Structural Similarity Index (SSIM), and (3) Standard Deviation of $\ell_2$-norm ($\ell_2$-STD). These metrics jointly capture attackability, perceptual imperceptibility, and optimization stability.

We evaluate on ImageNet, CIFAR-100, and MNIST datasets with diverse model architectures: ResNet50, Inception-v3, VGG19, ViT, WideResNet (WRN), Engstrom Wong & Kolter (2018), and Lipschitz Tsuzuku et al. (2018). For each model, we randomly sample 500 images from the test set. We adopt untargeted attacks with a query budget of 10,000 queries. Perturbation thresholds are set to $\epsilon = 2.5$ on ImageNet, $\epsilon = 1.0$ on CIFAR-100, and $\epsilon = 3.0$ on MNIST. We evaluate two base attackers, CGBA and ADBA, optionally enhanced with PAR or SeRI.

Table 8: Attack success rate (ASR) of SeRI measured under a query budget of 10,000.

| | Imagenet -ResNet50 | Imagenet -InceptionV3 | Imagenet -VGG19 | Imagenet -Engstrom | CIFAR100 -ViT | CIFAR100 -WRN | MNIST -Lipschitz |
|---|---|---|---|---|---|---|---|
| CGBA | 49.8% | 77.6% | 86.4% | 28.4% | 79.4% | 55.6% | 76.6% |
| CGBA+PAR | 56.0% | 82.8% | 87.2% | 38.8% | 84.6% | 58.4% | 82.8% |
| CGBA+SeRI | **64.6%** | **86.8%** | **91.0%** | 46.8% | **87.2%** | **61.8%** | **88.0%** |
| ADBA | 46.0% | 58.6% | 61.2% | 18.0% | 36.0% | 27.2% | 51.2% |
| ADBA+PAR | 52.6% | 62.8% | 71.0% | 41.0% | 47.8% | 36.6% | 72.6% |
| ADBA+SeRI | 57.8% | 66.2% | 79.4% | **49.8%** | 58.0% | 43.8% | 87.0% |

Table 9: Average Structural Similarity Index (SSIM) of SeRI on the ImageNet dataset, measured under a query budget of 10,000.

| | Imagenet -ResNet50 | Imagenet -InceptionV3 | Imagenet -VGG19 | Imagenet -Engstrom | CIFAR100 -ViT | CIFAR100 -WRN | MNIST -Lipschitz |
|---|---|---|---|---|---|---|---|
| CGBA | 0.959 | 0.982 | 0.994 | 0.903 | 0.996 | 0.964 | 0.449 |
| CGBA+PAR | 0.963 | 0.985 | **0.996** | 0.940 | **0.998** | 0.971 | 0.751 |
| CGBA+SeRI | 0.964 | **0.986** | **0.996** | 0.946 | **0.998** | **0.973** | **0.786** |
| ADBA | 0.962 | 0.972 | 0.982 | 0.903 | 0.981 | 0.910 | 0.445 |
| ADBA+PAR | 0.966 | 0.977 | 0.983 | 0.951 | 0.985 | 0.930 | 0.723 |
| ADBA+SeRI | **0.967** | 0.979 | 0.984 | **0.958** | 0.987 | 0.933 | 0.745 |

The results in Tables 8, 9, and 10 collectively highlight the clear and consistent advantages brought by SeRI across all datasets and model architectures. Most notably, SeRI provides a substantial improvement in attackability, achieving the highest ASR in every tested setting and outperforming both base attacker and PAR by large margins. This demonstrates that SeRI's sensitivity-guided

Table 10: STD of $\ell_2$-norm under a query budget of 10,000.

|  | Imagenet -ResNet50 | Imagenet -InceptionV3 | Imagenet -VGG19 | Imagenet -Engstrom | CIFAR100 -ViT | CIFAR100 -WRN | MNIST -Lipschitz |
|---|---|---|---|---|---|---|---|
| CGBA | 7.404 | 4.297 | 1.318 | 16.99 | 0.484 | 1.001 | 3.214 |
| CGBA+PAR | 5.972 | 3.661 | 1.265 | 4.891 | 0.432 | 0.933 | 1.997 |
| CGBA+SeRI | 4.636 | **2.398** | **1.124** | **3.628** | **0.390** | **0.853** | 0.939 |
| ADBA | 3.648 | 3.700 | 2.294 | 7.508 | 1.321 | 1.902 | 1.034 |
| ADBA+PAR | 3.211 | 3.505 | 2.007 | 4.659 | 1.165 | 1.451 | 0.871 |
| ADBA+SeRI | **2.789** | 3.448 | 1.856 | 3.653 | 1.032 | 1.126 | **0.738** |

refinement is highly effective in driving perturbations toward the true decision boundary even under strict query constraints.

At the same time, SeRI preserves exceptional perceptual quality, achieving SSIM scores that match or exceed both the base attackers and PAR, and often approaching values near 1.0, indicating that SeRI generates adversarial examples that remain visually indistinguishable from clean images.

Furthermore, SeRI consistently achieves the lowest $\ell_2$-STD, revealing that its continuous region-aware optimization yields far more stable and reliable perturbation magnitudes across images. Importantly, these improvements occur simultaneously: higher ASR, better perceptual quality, and greater optimization stability, demonstrating that SeRI offers a comprehensive enhancement rather than a trade-off. Overall, the results confirm that SeRI is a robust and broadly effective refinement module that significantly strengthens decision-based black-box attacks across diverse architectures and evaluation metrics, establishing it as a strong and practical advancement for the field.

