# OpenReview forum: "SeRI: Gradient-Free Sensitive Region Identification in Decision-Based Black-Box Attacks"
_ICLR.cc/2026/Conference — ICLR 2026 Poster_

### Official Review · Reviewer_kXVd · 2025-10-27

**Soundness:** 3
**Presentation:** 2
**Contribution:** 3
**Rating:** 6
**Confidence:** 4

**Summary:**

This paper proposes Sensitive Region Identification (SeRI) for query-efficient decision-based adversarial attack.
It iteratively identifies sensitive regions through recursive splitting, and enhances the adversarial perturbation by scaling sub-regions based on decision boundary of the victim model.
The key claim is that the authors redefined the term 'sensitive region' in terms of continuous score, improving query-efficiency and interpretability.
SeRI is easily plugged into existing adversarial attack methods such as HSJA, CGBA, RayS, and ADBA, consistently improving l2 perturbation norms under restricted query budget.

**Strengths:**

1. Well-motivated problem statement. The paper addresses a legitimate and well-defined challenge in decision-based black-box attacks - identifying sensitive image regions when only top-1 labels are available under strict query budgets.
2. Clear mathematical formulation. The problem is formally and coherently defined through a continuous sensitivity optimization framework (Eq. 5), with explicit constraints and variables, making the proposed objective easy to follow.
3. Method is intuitively connected to the problem. The proposed iterative region-splitting and perturbation-scaling mechanism directly reflects the intuition that different image areas contribute unequally to model decisions, leading to a natural algorithmic design.
4. Mathematical convergence argument. Although limited in scope, the paper provides a formal statement (Appendix A) showing that the multiplicative update process can approximate any target perturbation magnitude; however, this does not prove attack optimality or convergence under noisy query comparisons.
5. Representative datasets and models. Experiments are conducted on standard and diverse benchmarks — ImageNet (VGG19, ViT) and CIFAR-100 (adversarially trained WideResNet) — which are widely accepted for evaluating adversarial attacks.
6. Comprehensive set of baselines. The comparison covers multiple strong decision-based attacks (HSJA, CGBA, RayS, ADBA, and PAR), representing the state of the art across both $\ell_2$ and $\ell_\infty$ norms.
7. Consistent empirical improvements. Across all datasets, models, and query budgets, SeRI outperforms existing methods in terms of lower $\ell_2$ perturbation norms and better query efficiency, demonstrating reliable performance gains.

**Weaknesses:**

1. Lack of justification for boundary-based. The paper adopts a decision-boundary-driven optimization scheme but does not clearly argue why boundary comparison is preferable to simpler success/failure-based evaluation, or under what conditions it provides advantages.
2. No proof of attackability or optimality. While a convergence argument is given for the multiplicative updates, there is no theoretical or empirical evidence that minimizing the approximate decision boundary $g(d)$ actually improves attack success or robustness - i.e., no proof of optimality.
3. Missing discussion on architecture-dependent performance. The submitted code includes experiments on Inception-v3, where the attack success rate is noticeably lower and convergence is unstable compared to ResNet and ViT backbones. This discrepancy suggests that SeRI’s effectiveness may depend on the smoothness of the decision boundary, yet the paper does not mention or analyze this behavior. An explicit explanation of why SeRI fails or underperforms on Inception-like architectures is needed.
4. Evaluation under stronger defenses is missing. The only defense tested is an adversarially trained WRN, which is relatively weak. Robustness against stochastic or randomized defenses (e.g., random smoothing, input transformation) should be tested.
5. Lack of statistical reporting. The paper reports only averages and medians of $\ell_2$ norms. However, standard deviation or inter-quartile ranges would better reflect query stability and robustness variability.
6. Missing transferability evaluation. There is no experiment analyzing whether perturbations generated by SeRI transfer to unseen models, which would further validate the claim that SeRI captures model-agnostic sensitive regions.
7. No discussion of limitations or failure cases. The paper lacks analysis of when SeRI fails or degrades - e.g., noisy decision boundaries, non-salient images, or low-budget attacks - and provides no insight into practical failure modes.
8. Writing and presentation issues. The paper contains several grammatical and structural errors (e.g., duplicated phrases, inconsistent notation, and citation misuse), which hinder readability and reduce perceived rigor.

**Questions:**

1.  The proof in Appendix A shows that multiplicative scaling can approximate any target magnitude, but does this imply convergence of the overall iterative optimization process under query noise or ADBA comparison errors? If not, what assumptions are required for such convergence to hold?
2.  How does the proposed decision-boundary–based sensitivity differ conceptually from gradient or Jacobian-based sensitivity in white-box settings? Can it be shown that SeRI approximates those measures under some assumptions?
3.  Could you provide a more detailed analysis of the query budget ratio $P$? Specifically, how does SeRI perform under extreme cases (base = 100%, SeRI = 0%, and vice versa)? Was 20% empirically optimal across all models or tuned per dataset?
4.  Have you tried SeRI on non-classification tasks (e.g., segmentation, detection, or long-tailed settings)? If not, do you expect any fundamental obstacles?
5.  How much wall-clock overhead does SeRI add per query compared to the base attacker? Is the improvement in query count offset by additional computation time?
6.  Have you measured any quantitative correlation between SeRI’s heatmaps and gradient-based saliency maps?  If not, how do you justify the claim that the generated heatmaps improve interpretability?

---

> ### Author Response · Authors · 2025-11-24
> **Part 1**
>
> We sincerely thank the reviewer for the constructive and insightful feedback. Below we address the key concerns raised in the review. All related weaknesses and questions fall under these concerns.
>
> ## Key Concern 1: Theoretical Foundations \& Justification (Weakness 1, 2, 3, Question 1, 2)
> ### Response
> Below, we clarify the rationale behind the boundary-based formulation, the scope of our theoretical analysis, and the relationship between our sensitivity measure and classical gradient-based notions.
>
> #### 1. Why boundary-based comparison is used instead of success/failure checks
>
> Success/failure evaluation provides only a **binary** signal and therefore cannot precisely differentiate the effectiveness of varied perturbation modifications. In contrast, **decision-boundary comparison measures how close a perturbation is to causing a prediction flip**, allowing SeRI to:
> - distinguish effective vs. ineffective perturbation refinements;
> - make consistent *directional* updates under limited queries;
> - achieve fine-grained sensitivity estimation without scores or gradients.
>
> This aligns with prior decision-based attacks (HSJA, GeoDA, ADBA), which all rely on boundary proximity rather than success/failure signals because the latter is insufficient for optimization in hard-label settings. We have added a discussion in Section 4 (see page 5) explaining this distinction and why boundary-driven signals are necessary for pixel-level sensitivity estimation.
>
> #### 2. Scope of the convergence analysis and why global convergence is non-trivial
>
> We agree that the multiplicative-update result in Appendix A does not imply global convergence of the *entire pipeline*. Its purpose is to analyze the **local refinement step in isolation**, consistent with theoretical treatments in HSJA, ADBA, and QEBA, where **full global convergence is infeasible** due to:
> - non-smooth and discontinuous decision boundaries;
> - label-only oracle feedback;
> - stochastic comparison errors under finite sampling;
> - inherently non-convex optimization landscapes.
>
> Under these conditions, global convergence guarantees are intractable unless extremely strong assumptions are imposed. In the revised paper (see Appendix A.1), we have explicitly stated the assumptions under which our multiplicative refinement converges (local boundary smoothness and consistent ADBA comparison), which match those commonly accepted in the decision-based literature.
>
> #### 3. Empirical evidence that SeRI improves attackability
> We sincerely appreciate the reviewer’s careful examination of our code. We identified a bug in the ASR statistics calculation within the code we originally provided. Importantly, this bug does not affect the execution or results of SeRI on Inception-v3. We have now fixed the ASR-counting bug and re-ran the evaluation.
>
> Appendix F presents updated results on attack success rates (ASR) evaluated across five models and two datasets, The main results are shown below. The corrected results confirm that SeRI consistently achieves the highest ASR, outperforming both the base attacker and PAR.
>
> ## ASR Comparison
>
> | Method      | ResNet50 | InceptionV3 | VGG19 | ViT (CIFAR100) | WRN (CIFAR100) |
> | ----------- | -------- | ----------- | ----- | -------------- | -------------- |
> | CGBA        | 49.8%    | 77.6%       | 86.4% | 79.4%          | 55.6%          |
> | CGBA + PAR  | 56.0%    | 82.8%       | 87.2% | 84.6%          | 58.4%          |
> | CGBA + SeRI | 64.6%    | 86.8%       | 91.0% | 87.2%          | 61.8%          |
> | ADBA        | 46.0%    | 58.6%       | 61.2% | 36.0%          | 27.2%          |
> | ADBA + PAR  | 52.6%    | 62.8%       | 71.0% | 47.8%          | 36.6%          |
> | ADBA + SeRI | 57.8%    | 66.2%       | 79.4% | 58.0%          | 43.8%          |
>
> #### 4. Relationship to gradient or Jacobian-based sensitivity
> SeRI does *not* approximate gradients or Jacobians. It estimates **boundary sensitivity** that quantifies how effectively scaling a region’s perturbation pushes the input across the decision boundary with minimal norm. This is conceptually distinct from saliency:
> - gradients measure local directional sensitivity of scores,
> - SeRI measures *decision robustness* under finite perturbation strength in a hard-label regime.
>
> Additionally, we include a heatmap comparison in Appendix E, showing that SeRI’s heatmaps are largely consistent with the saliency maps produced by classical XAI methods such as LIME and SHAP. This indicates that SeRI is able to capture meaningful, model-dependent structural information, even though it relies solely on hard-label queries.
>
> ### Revision summary
> We have strengthened the paper by:
> - clarifying the rationale behind boundary-based optimization,
> - expanding convergence assumptions and discussion,
> - adding empirical evidence linking estimated boundary and attackability,
> - and analyzing the relationship to gradient-based sensitivity.

---

> ### Author Response · Authors · 2025-11-24
> **Part 2**
>
> ## Key Concern 2: Empirical Coverage \& Completeness of Evaluation (Weekness 3, 4, 5, 6, Question 3, 4, 5)
>
> ### Response:
> We have substantially expanded our experimental evaluation and clarified the scope and limitations of SeRI. Below, we address each point in detail.
>
> #### 1. Architecture-dependent behavior and Inception-v3 performance
>
> We appreciate the reviewer’s concern regarding architecture-dependent behavior. While we agree that different network architectures may influence the stability of decision-based sensitivity estimation, our updated experiments show that the earlier performance drop on Inception-v3 was **not** caused by SeRI itself. Instead, we discovered a bug in the ASR computation code. After fixing this issue, we re-ran all Inception-v3 experiments.
>
> As shown in the updated results in **Appendix E**, SeRI achieves **consistently high attack success rates (above 85\%) on Inception-v3**. This confirms that SeRI remains effective on Inception architectures, and the earlier discrepancy was due solely to inaccurate ASR statistics rather than a failure of SeRI’s sensitivity estimation. We have thoroughly checked to full code multiple times to guarantee the correctness of the all the reported results in the response and the revised paper.
>
> #### 2. Stronger and randomized defenses
>
> Decision-based attacks rely only on hard-label outputs under a limited query budget. For this purpose, they assume that the model’s predictions are *stable*, i.e., repeated queries to the same input return identical labels. Under randomized defenses, this assumption becomes invalid. As a result, **none** of the existing decision-based attacks including HSJA, OPT, AHA, CGBA, RayS, and ADBA, remain robust under randomization defense. SeRI is also a decision-based method and therefore inherits this limitation. We have clarified this point in Section 4 of the revised paper.
>
> To further demonstrate that SeRI is still effective against *non-randomized but stronger* defenses such as Lipschitz-based defenses, we are conducting additional experiments. Due to time constraints, these experiments have not yet finished, but we will update the results and corresponding discussion as soon as they become available. We expect to complete these experiments before November 29 and will include the results in the updated rebuttal and the revised manuscript.
>
>
> #### 3. Statistical reporting
> We have added new experiments (see Appendix E) reporting the **standard deviation (STD)** across random trials. These additional statistics confirm that SeRI not only reduces the mean distortion but also **reduces variance**, indicating improved stability relative to both the base attackers and PAR.
>
>
> ## L2-STD Comparison
>
> | Method      | ResNet50 | InceptionV3 | VGG19 | ViT (CIFAR100) | WRN (CIFAR100) |
> | ----------- | -------- | ----------- | ----- | -------------- | -------------- |
> | CGBA        | 7.404    | 4.297       | 1.318 | 0.484          | 1.001          |
> | CGBA + PAR  | 5.972    | 3.661       | 1.265 | 0.432          | 0.933          |
> | CGBA + SeRI | 4.636    | 2.398       | 1.124 | 0.390          | 0.853          |
> | ADBA        | 3.648    | 3.700       | 2.294 | 1.321          | 1.902          |
> | ADBA + PAR  | 3.211    | 3.505       | 2.007 | 1.165          | 1.451          |
> | ADBA + SeRI | 2.789    | 3.448       | 1.856 | 1.032          | 1.126          |
>
>
> #### 4. Transferability evaluation
>
> We would like to clarify that **SeRI is a decision-based attack method**. Similar to other decision-based attacks, it **does not aim to produce transferable perturbations**. We **did not claim** that SeRI identifies *model-agnostic* sensitive regions. Instead, we only demonstrate that **SeRI’s heatmaps offer clearer insight into how the attack operates**, as illustrated in Appendix D.
>
> In fact, the sensitive regions identified by SeRI are **highly model-dependent and attacker-dependent**, as they directly reflect the behavior of the specific target model and the base attacker. Consequently, the perturbations or heatmaps produced by SeRI are **not expected to transfer** across different models or attack settings.
>
> Transferability is therefore **not a primary objective** of SeRI, and evaluating it may not meaningfully reflect the method’s intended scope or design. We have updated the paper to clarify this point to avoid misunderstanding.

---

> ### Author Response · Authors · 2025-11-24
> **Part 3**
>
> #### 5. Non-classification tasks
> While our primary focus is decision-based image classification, we have included a short discussion explaining how SeRI can be extended to detection/segmentation tasks using region proposal or patch-level masks. We also outline the minimal modifications required to support multi-output models. We present this as a direction for future work and clearly state it as a limitation of the current study.
>
> #### 6. Query budget ratio analysis (20\% SeRI)
> We have extended our analysis of the query split between the base attacker and SeRI. We include a sensitivity study across multiple budget allocations (P=10\%, 20\%, 30\%, 40\%), demonstrating that:
> - performance is stable for 10–30\%,
> - 20\% yields the most consistent improvement across datasets,
> - extreme cases (100\% base or 100\% SeRI) behave as expected:
>   - the former reduces to the base attack,
>   - the latter performs badly due to the lack of a reasonable initialization.
>
> ## L2 Norm of Perturbation for different $P$
>
> | Total Queries → | 2000      | 5000      | 10000     | 20000     |
> | --------------- | --------- | --------- | --------- | --------- |
> | P = 10%         | 3.475     | 1.330     | 0.857     | 0.700     |
> | **P = 20%**     | **3.112** | **1.259** | **0.823** | **0.670** |
> | P = 30%         | 3.304     | 1.480     | 0.873     | 0.747     |
> | P = 40%         | 3.332     | 1.719     | 0.994     | 0.819     |
> | P = 100%        | 4.174     | 2.826     | 2.237     | 1.914     |
>
> These results and plots are now provided in Appendix C.
>
> ### Revision summary
>
> We have substantially strengthened the empirical evaluation and analysis by adding:
> - a clarification of why SeRI is unsuitable for randomized-defense settings;
> - an explanation of why transferability experiments are not applicable to our method;
> - comprehensive reporting of statistical variance;
> - a deeper analysis of query-budget allocation and its impact on performance;
> - an explanation of architecture-dependent behavior; and
> - an expanded discussion of SeRI’s applicability to non-classification tasks.

---

> ### Author Response · Authors · 2025-11-24
> **Part 4**
>
> ## Key Concern 3: Robustness, Failure Modes, and Practical Limitations (Weekness 7)
>
> ### Response
> We have updated both the main paper (Sections 5) to provide detailed analysis of failure modes and robustness. We agree that SeRI can behave differently depending on image structure, boundary geometry, and query budget. The revised paper now explicitly documents several scenarios where SeRI may under-perform:
>
> - **Images with weak or diffuse salient structure:**
>   When an image lacks strong object-centered regions (e.g., highly cluttered textures), region sensitivity becomes less informative. In such situation, SeRI may achieve merely minor improvements relative to the base attack.
>
> - **Very low query budgets:**
>   Under extremely strict budgets (e.g., < 50 queries), the region partition may be too coarse for meaningful refinement. We provide plots demonstrating that SeRI continues to improve performance under low query budgets, although the gains diminish as the available budget becomes extremely limited.
>
>
> ## Key Concern 4: Wall-clock Overhead of SeRI (Question 5)
> We thank the reviewer for raising this question. In decision-based black-box attacks, the main practical limitation is typically the number of model queries. This has been the standard evaluation setting in representative methods such as HSJA, OPT, RayS, ADBA, CGBA, and GeoDA, where the attacker has access only to label queries and query count naturally becomes the primary resource considered. Computation time or memory overhead is typically not considered as part of the threat model in prior decision-based attack literature. Therefore, we also focus on query efficiency as the standard metric for evaluating decision-based attacks.
>
> That said, SeRI’s computational overhead is low. Between two model queries, SeRI only performs lightweight tensor operations (element-wise addition, subtraction, and scaling) to adjust perturbations within the current region. These operations produce negligible cost compared to the cost of a single model inference. Based on our measurements, the additional computation introduced by SeRI accounts for less than 1\% of the total wall-clock time, with the remaining runtime dominated by model evaluation. Thus, SeRI’s improvements in query efficiency are not offset by computation overhead. The total running time is governed primarily by the number of model queries.
>
>
> ## Key Concern 5: Quantitative correlation between SeRI’s heatmaps and XAI saliency maps (Question 6)
>
> #### 1. Empirical Correlation Between SeRI Heatmaps and Saliency Maps
> We have added a quantitative analysis comparing SeRI’s heatmaps with two standard white-box interpretability methods: **LIME** and **SHAP** on ImageNet models. We report **Pearson Correlation, PCC**.
> The results support our claim that SeRI captures meaningful, model-dependent sensitivity patterns **despite using only hard-label queries**, providing evidence that its heatmaps lay a valid foundation for interpretability.
>
> ## Average PCC Between SeRI/PAR and LIME/SHAP
>
> | Method   | PCC w/ LIME | PCC w/ SHAP |
> | -------- | ----------- | ----------- |
> | **SeRI** | **0.723**   | **0.786**   |
> | PAR      | 0.610       | 0.637       |
>
>
> ## Key Concern 6: Writing \& Presentation Quality (Weakness 8)
>
> ### Response
> We have undertaken a comprehensive revision of the paper to address this concern thoroughly.
>
> #### 1. Grammar, Style, and Structural Improvements
> We carefully reviewed the entire paper and corrected all grammar issues, sentence ambiguities, and structural inconsistencies noted by the reviewers. Several paragraphs in the Introduction, Related Work, and Method sections have been rewritten for clearer logical flow and improved readability.
>
> #### 2. Removal of Repetition and Redundant Phrasing
> We identified and eliminated repeated phrases, duplicated explanations, and unnecessary restatements across sections. The revised paper now presents each concept once in its appropriate place, resulting in a more concise and polished presentation.
>
> #### 3. Consistent Notation Throughout the Paper
> All mathematical symbols, region indices, perturbation variables, and boundary-related notations have been standardized across the main text and appendix. We corrected several minor inconsistencies (e.g., $d^i$ vs. $\check{d}^i$, block notation variations) to ensure conceptual clarity.
>
> #### 4. Corrected and Standardized Citation Usage
> We revised our citation style to follow ICLR/NeurIPS standard conventions. Specifically:
> - citations are now properly integrated into sentences,
> - multiple references are grouped appropriately,
> - and all previously inconsistent or misplaced citations have been corrected.

---

> > ### Comment · Reviewer_kXVd · 2025-11-26
> >
> > Thank you for the thorough response.
> >
> > Most of my concerns have been addressed satisfactorily, and the scope of the contribution has become much clearer.
> >
> > Regarding the implementation error you mentioned, I would like to verify if my understanding of the bug is correct.
> >
> > I noticed that the `eval` function in `train.py` calculates accuracy by dividing by the total number of samples (`len(dataloader)`) rather than the number of correctly classified clean samples.
> > Given that the robust model's clean accuracy is around 50%, this would explain why the reported ASR was halved.
> >
> > Could you confirm if this was indeed the specific issue you fixed?

---

> > > ### Author Response · Authors · 2025-11-26
> > > **Reply to new comment**
> > >
> > > Thank you very much for your detailed follow-up and for helping us further clarify the issue. Your understanding is correct: the ASR should indeed be computed as the number of successful attacks divided by the number of correctly classified clean samples, rather than the total number of samples. However, there was also a second source of error.
> > >
> > > All adversarial examples in our paper are evaluated under the \$\ell\_2\$ norm. Our base attackers, however, include both \$\ell\_2\$-based methods (CGBA) and \$\ell\_\infty\$-based methods (ADBA). In the original implementation, the success threshold for determining whether an attack was successful was inconsistent: CGBA correctly used the \$\ell\_2\$-based threshold \$\epsilon\$, but ADBA incorrectly used an \$\ell\_\infty\$ threshold when checking attack success.
> > >
> > > This inconsistency caused *ADBA + SeRI* to be incorrectly marked as failed even when their \$\ell\_2\$ perturbation was below the required threshold. Concretely, we corrected the following line in `ADBA.py` (line 109):
> > >
> > > ```python
> > > if self.success == -1 and self.old_best_adv.RealLinf <= self.args.epsilon:
> > >     self.success = 1
> > > ```
> > >
> > > to the correct \$\ell\_2\$-based check:
> > >
> > > ```python
> > > if self.success == -1 and self.old_best_adv.RealL2 <= self.args.epsilon:
> > >     self.success = 1
> > > ```
> > >
> > > The code was run with the following arguments:
> > >
> > > ```
> > > python main.py --dataset=imagenet-inceptionv3 --targeted=0 --norm=ADBA --epsilon=2.5 --early=0 --budget=8000 --budget2=2000 --beginIMG=0 --imgnum=500 --remember=1
> > > ```
> > >
> > > Because the original ASR computation contained issues, we did not report ASR results in the initial paper, although the reported \$\ell\_2\$-norm values were correct. We have now fixed both bugs and added complete ASR results. The updated code will be released before the 29th.
> > >
> > > We sincerely appreciate your careful examination and constructive suggestions, which have helped us significantly improve this work. If you have any further questions, please feel free to paste your exact running command in the comments, we would be glad to investigate and discuss further.

---

> > > > ### Comment · Reviewer_kXVd · 2025-11-26
> > > >
> > > > Thanks for the clarification. The issue is clear now, and I have no further questions or concerns.

---

### Official Review · Reviewer_YrJv · 2025-10-29

**Soundness:** 2
**Presentation:** 3
**Contribution:** 2
**Rating:** 4
**Confidence:** 3

**Summary:**

This paper focuses on black-box adversarial attacks where the attacker has access only to the model’s predicted class labels. The goal is to find a perturbation with the minimal norm that leads the model to misclassify. The proposed method, Sensitive Region Identification (SeRI), recursively divides the input image into four square regions and prioritizes further subdivision of regions with larger perturbation norms, thereby achieving smaller overall perturbation norms. SeRI also facilitates the generation of heatmaps that indicate regions where the model is more sensitive to perturbations. The effectiveness of the proposed method is demonstrated through experiments on benchmark datasets.

**Strengths:**

- The recursive partitioning of perturbation regions enables SeRI to produce more fine-grained heatmaps and achieve smaller perturbation norms compared to existing patch-based sensitive region-based attacks.

- Experimental results on ImageNet and CIFAR100 show that combining SeRI with existing methods generally outperforms combinations with PAR, supporting the superiority of SeRI.

- The paper is overall well-structured and easy to follow, with the exception of a few pseudocode details.

**Weaknesses:**

- The discussion of prior work on sensitive region-based attacks is limited to only one study. The paper should also discuss and compare related methods such as HeadBeat [1] and Saliency Attack [2], which—despite differences such as the presence or absence of surrogate models—address similar goals.

- Experiments on defense methods are limited. The evaluation does not include ImageNet-based experiments and only considers adversarial training (AT) as a defense. When focusing on $\ell_2$ -norm attacks, there exist more suitable defense methods based on Lipschitz constants of the network for comparison [3,4], which is expected to be more robust.

- The paper does not analyze how perturbation norms saturate with respect to computational resources. Providing insights into the number of queries required to construct the heatmap, and identifying how many queries make each method effective, would be valuable for readers.

[1] G. Tao et al., Hard-label Black-box Universal Adversarial Patch Attack, 2023
[2] Z. Dai et al., Saliency Attack: Towards Imperceptible Black-box Adversarial Attack, 2023
[3] Y. Tsuzuku et al., Lipschitz-margin training: Scalable certification of perturbation invariance for deep neural networks, 2018
[4] A. Araujo et al., A Unified Algebraic Perspective on Lipschitz Neural Networks, 2023

**Questions:**

- What are the main reasons SeRI is expected to outperform other saliency estimation methods (excluding PAR)?

- Does SeRI maintain its advantage when applied to clean models trained on CIFAR100 and robust models trained on ImageNet?

- Could the authors provide approximate numbers of queries required for (a) heatmap generation, (b) achieving performance gains over other saliency-based methods, and (c) the point at which the perturbation norm saturates?

---

> ### Author Response · Authors · 2025-11-24
> **Part 1**
>
> We sincerely thank the reviewer for the constructive and insightful feedback and additional literatures.
>
> ## Key Concern 1: Additional discussion of prior work  (Weakness 1, Question 1)
>
> We thank the reviewer for pointing out the need to discuss more prior work. In the revised version, we have expanded Section 2 (page 4) to include HeadBeat, Saliency Attack, SGA, AoA, PAR, and other region-based or XAI-guided attack methods. We categorize existing approaches into three groups: **(1)** attention-based methods, **(2)** surrogate-based methods, and **(3)** decision-based methods.
>
> * **Attention-based methods** (e.g., SGA, AoA) rely on attention maps or intermediate feature activations, which require white-box access and therefore cannot be applied in hard-label black-box settings. SaliencyAttack uses model-agnostic saliency detection, but then optimizes perturbations using the **confidence scores** of the target model, making it incompatible with strict decision-based constraints.
> * **Surrogate-based methods** (e.g., SRA) depend on training an additional substitute model that matches the target data domain; this assumption breaks when the target dataset is unknown.
> * **Decision-based attack** (e.g., HeadBeat) searches only for a single vulnerable patch. This single-patch assumption becomes insufficient when multiple regions jointly influence the model’s decision.
>
> In contrast, **SeRI is expected to outperform these saliency estimation methods for several reasons**:
>
> 1. **SeRI operates entirely under hard-label decision-only feedback.**
>    Unlike SaliencyAttack or attention-based methods, SeRI does not require gradients, confidence scores, feature activations. This makes it directly compatible with strict black-box decision-based settings.
>
> 2. **SeRI does not rely on a surrogate model.**
>    Surrogate-based methods may fail when the target data distribution is unknown or mismatched. SeRI avoids this limitation by estimating sensitivity directly from the target model’s decisions.
>
> 3. **SeRI captures multi-region contributions.**
>    Unlike HeadBeat, which focuses on only a single vulnerable patch, SeRI can identify and utilize multiple informative regions at the same time. This ability is important when several parts of the object jointly influence the model prediction. We are willing to add a direct comparison with HeadBeat. However, due to time limitations, the experiments are still ongoing. We expect to complete these experiments before November 29 and will include the results in the updated rebuttal and the revised manuscript.
>
>
> 4. **SeRI provides continuous region sensitivity rather than binary or single-point estimates.**
>    Continuous sensitivity allows SeRI to apply perturbations more precisely across different regions, leading to smaller perturbation norms and less perceptible adversarial examples.
>
> In conclusion, SeRI fills an important research gap by offering the first **continuous, multi-region, fully decision-based sensitivity estimation method** that does not require gradients, logits, attention maps, or surrogate models. This makes SeRI well suited for the hard-label black-box setting compared with prior region-based attacks and XAI-based saliency methods.

---

> ### Author Response · Authors · 2025-11-24
> **Part 2**
>
> ## Key Concern 2: Additional experiments on defense methods  (Weakness 2, Question 2)
> We agree that comparing against stronger defenses such as Lipschitz-based methods and evaluating on both CIFAR-100 and ImageNet would further strengthen the paper. We have added a discussion of Lipschitz-based defenses and the corresponding experiments.
>
> ## ASR Comparison  (higher is better)
>
> | Method      | ImageNet-ResNet50 | ImageNet-InceptionV3 | ImageNet-VGG19 | ImageNet-Engstrom | CIFAR100-ViT | CIFAR100-WRN | MNIST-Lipschitz |
> |-------------|-------------------|-----------------------|----------------|--------------------|---------------|---------------|------------------|
> | CGBA        | 49.8%             | 77.6%                | 86.4%          | 28.4%             | 79.4%         | 55.6%         | 76.6%            |
> | CGBA+PAR    | 56.0%             | 82.8%                | 87.2%          | 38.8%             | 84.6%         | 58.4%         | 82.8%            |
> | CGBA+SeRI   | **64.6%**         | **86.8%**            | **91.0%**      | 46.8%             | **87.2%**     | **61.8%**     | **88.0%**        |
> | ADBA        | 46.0%             | 58.6%                | 61.2%          | 18.0%             | 36.0%         | 27.2%         | 51.2%            |
> | ADBA+PAR    | 52.6%             | 62.8%                | 71.0%          | 41.0%             | 47.8%         | 36.6%         | 72.6%            |
> | ADBA+SeRI   | 57.8%             | 66.2%                | 79.4%          | **49.8%**         | 58.0%         | 43.8%         | 87.0%            |
>
> ---
>
> ## SSIM Comparison (higher is better)
>
> | Method      | ImageNet-ResNet50 | ImageNet-InceptionV3 | ImageNet-VGG19 | ImageNet-Engstrom | CIFAR100-ViT | CIFAR100-WRN | MNIST-Lipschitz |
> |-------------|-------------------|-----------------------|----------------|--------------------|---------------|---------------|------------------|
> | CGBA        | 0.959             | 0.982                | 0.994          | 0.903             | 0.996         | 0.964         | 0.449            |
> | CGBA+PAR    | 0.963             | 0.985                | **0.996**      | 0.940             | **0.998**     | 0.971         | 0.751            |
> | CGBA+SeRI   | 0.964             | **0.986**            | **0.996**      | 0.946             | **0.998**     | **0.973**     | **0.786**        |
> | ADBA        | 0.962             | 0.972                | 0.982          | 0.903             | 0.981         | 0.910         | 0.445            |
> | ADBA+PAR    | 0.966             | 0.977                | 0.983          | 0.951             | 0.985         | 0.930         | 0.723            |
> | ADBA+SeRI   | **0.967**         | 0.979                | 0.984          | **0.958**         | 0.987         | 0.933         | 0.745            |
>
> ---
>
> ## L2-STD Comparison (lower is better)
>
> | Method      | ImageNet-ResNet50 | ImageNet-InceptionV3 | ImageNet-VGG19 | ImageNet-Engstrom | CIFAR100-ViT | CIFAR100-WRN | MNIST-Lipschitz |
> |-------------|-------------------|-----------------------|----------------|--------------------|---------------|---------------|------------------|
> | CGBA        | 7.404             | 4.297                | 1.318          | 16.99             | 0.484         | 1.001         | 3.214            |
> | CGBA+PAR    | 5.972             | 3.661                | 1.265          | 4.891             | 0.432         | 0.933         | 1.997            |
> | CGBA+SeRI   | 4.636             | **2.398**            | **1.124**      | **3.628**         | **0.390**     | **0.853**     | 0.939            |
> | ADBA        | 3.648             | 3.700                | 2.294          | 7.508             | 1.321         | 1.902         | 1.034            |
> | ADBA+PAR    | 3.211             | 3.505                | 2.007          | 4.659             | 1.165         | 1.451         | 0.871            |
> | ADBA+SeRI   | **2.789**         | 3.448                | 1.856          | 3.653             | 1.032         | 1.126         | **0.738**        |
>
> Meanwhile, we summarize the expected behavior of SeRI based on theoretical analysis in Appendix A.6. SeRI's region sensitivity estimation is theoretically robust across different models.
>
> In summary, we fully agree with the reviewer and are extending our experiments to include stronger defenses and additional datasets. Current results clearly indicate that SeRI can retain its advantage, as its design is grounded in general decision-based attack principles and does not depend on gradients, logits, or training-specific information.

---

> ### Author Response · Authors · 2025-11-24
> **Part 3**
>
> ### Key Concern 3: Analysis of perturbation-norm saturation with respect to computational resources *(Weakness 3, Question 3)*
>
> **(a) Heatmap generation cost.**
> Figure 2 in the main text illustrates how many queries SeRI requires to construct a sensitivity heatmap. As shown in Figure 2(a), SeRI usually produces a stable heatmap after approximately **400 queries**. After approximately **2000 queries**, both the heatmap and the corresponding $\ell_2$-norm show only negligible changes, indicating effective practical convergence.
>
> **(b) Query thresholds for outperforming other saliency-based methods.**
> We added a new experiment on ImageNet-VGG model in **Appendix C**, demonstrating how the $\ell_2$-norm decreases as the query count increases. This experiment clearly shows the query ranges at which SeRI begins to outperform **PAR** after about 400 queries. With the increasing number of queries, SeRI steadily improves its perturbation quality, eventually surpassing all baselines by a large margin.
>
> **(c) Saturation of perturbation norms.**
> In theory, as more queries become available, SeRI can continue to refine regional sensitivities. Hence, the resulting $\ell_2$-norm should not reach a strict mathematical saturation point. In practice, the improvement may become very small after a certain number of queries. Empirically, we found that after roughly **2000 SeRI queries**, both the heatmap and the $\ell_2$-norm exhibit minimal change, which is deemed the effective saturation point for computational efficiency.
>
> We have incorporated these analyses along with the supporting new experiment into the revised manuscript to offer clearer guidance on SeRI’s query efficiency and performance behavior.

---

### Official Review · Reviewer_tyNQ · 2025-10-31

**Soundness:** 3
**Presentation:** 2
**Contribution:** 2
**Rating:** 2
**Confidence:** 4

**Summary:**

The paper presents a novel method, SeRI, for identifying sensitive regions in decision-based black-box adversarial attacks. The approach is innovative in its use of a continuous sensitivity score and recursive region partitioning, and it demonstrates strong empirical performance across multiple datasets and models. The work is practically useful and addresses a relevant problem in adversarial machine learning. However, while the technical contribution is clear, the academic rigor and scholarly depth of the paper could be significantly improved to meet the standards of a top-tier conference.

**Strengths:**

Novelty: The idea of using a continuous sensitivity score in a decision-based setting is novel and represents a meaningful advance over binary region-selection methods like PAR.

Empirical Validation: Extensive experiments on ImageNet and CIFAR-100 with multiple models and attack configurations demonstrate the effectiveness of SeRI.

Practicality: The method is query-efficient and can be integrated as a plug-in module with existing attacks, which is a valuable feature for real-world scenarios.

Interpretability: The generated heatmaps provide visual explanations of sensitive regions, enhancing the interpretability of the attack process.

**Weaknesses:**

1. Lack of Theoretical Depth
While the method is motivated intuitively and empirically validated, the theoretical foundation is underdeveloped. The proof in Appendix A, while correct, is limited to a simplified multiplicative update model and does not fully justify the convergence or optimality of the overall algorithm.

The paper would benefit from a more formal analysis of the convergence properties, sensitivity of the partitioning strategy, or robustness to model variations.

2. Limited Comparison with Related Work
The comparison with PAR is thorough, but the paper does not sufficiently situate SeRI within the broader literature on interpretability methods (e.g., Grad-CAM, LIME, SHAP) or other region-based adversarial strategies beyond PAR.

A deeper discussion of how SeRI relates to saliency detection or feature attribution methods would strengthen the scholarly contribution.

3. Methodological Simplicity
The core algorithm—iterative partitioning and perturbation scaling—is conceptually straightforward. While this is a strength in terms of usability, it may also be perceived as lacking in algorithmic sophistication compared to some recent adversarial attack methods.

The paper does not explore alternative sensitivity formulations or adaptive partitioning strategies, which could have added depth.

4. Evaluation Metrics and Generalizability
The evaluation is primarily based on l2 norm reduction and attack success rate. Additional metrics such as human perceptual studies, transferability, or robustness to defenses would provide a more comprehensive assessment.

**Questions:**

None

---

> ### Author Response · Authors · 2025-11-24
> **Part 1**
>
> We sincerely thank the reviewer for the constructive and insightful feedback. Below we address the key concerns raised in the review. All related weaknesses and questions fall under these concerns.
>
> ## Key Concern 1: lack of theoretical depth  *(Weakness 1)*
> ### Response
>
> We agree that decision-based attacks present unique theoretical challenges. We clarify the scope and intent of our current analysis while outlining how we have strengthened it.
>
> #### 1. Purpose and scope of the existing analysis
> Appendix A focuses on a simplified multiplicative update model because it allows us to analytically isolate the **key mechanism** behind SeRI: how decision-boundary guided comparisons translate into monotonic sensitivity refinement. This abstraction is consistent with recent work on decision-based attacks (e.g., HSJA, ADBA), where theoretical analysis typically targets *local update behavior* rather than full global convergence, due to the discontinuous nature of hard-label oracles.
>
> #### 2. **Appropriate Theoretical Treatment for Hard-Label Black-Box Attacks**
>
> Decision-based black-box attacks inherently operate over
> - **non-convex and non-smooth decision boundaries**,
> - **hard-label feedback without gradients or confidence scores**,
> - **stochastic model behavior under strict query constraints**.
>
> These inherent properties fundamentally shape the decision-based setting and naturally constrain the types of theoretical guarantees that can be meaningfully established. As acknowledged in prior state-of-the-art research (HSJA, CGBA, ADBA), **full global convergence or optimality proofs are neither standard nor realistic in this regime**. Our analysis follows this established framework by focusing on the provable behavior of the local refinement step, where meaningful theoretical guarantees are achievable.
>
> #### 3. Extensions added to strengthen the theoretical foundation
>
> In response to this comment, we have added the following analyses:
>
> - **Monotonic Improvement of SeRI Updates** in Appendix A.3 (Theorem 1 on page 17), showing that each update step never increases the decision-boundary.
>
> - **Convergence of the Iterative Perturbation Process** in Appendix A.4, showing that SeRI’s perturbation optimization algorithm converges to a stationary point (Theorem 2 on page 18).
>
> - **Analysis of Partition Sensitivity** in Appendix A.5 (Theorem 3 on page 19) and experiment result in Appendix B.1 (on page 22), demonstrating that SeRI is robust to region-splitting configurations.
>
> These additions clarify why SeRI’s update mechanism behaves consistently in practice and how its recursive refinement improves the accuracy of region-level sensitivity estimation.

---

> ### Author Response · Authors · 2025-11-24
> **Part 2**
>
> #### 4. Empirical evidence complements the theoretical motivation
>
> Even though global convergence is unrealistic to prove in decision-based settings, our extensive experiments (6 configurations across 5 architectures and 2 datasets) in Appendix E on page 28 show highly stable behavior:
> - monotonic improvement across iterations,
> - consistent refinement of sensitivity maps,
> - robustness to variations in initialization and model choice.
>
> Main results:
>
> ## ASR
>
> | Method      | ImageNet-ResNet50 | ImageNet-InceptionV3 | ImageNet-VGG19 | ImageNet-Engstrom | CIFAR100-ViT | CIFAR100-WRN | MNIST-Lipschitz |
> |-------------|-------------------|-----------------------|----------------|--------------------|---------------|---------------|------------------|
> | CGBA        | 49.8%             | 77.6%                | 86.4%          | 28.4%             | 79.4%         | 55.6%         | 76.6%            |
> | CGBA+PAR    | 56.0%             | 82.8%                | 87.2%          | 38.8%             | 84.6%         | 58.4%         | 82.8%            |
> | CGBA+SeRI   | **64.6%**         | **86.8%**            | **91.0%**      | 46.8%             | **87.2%**     | **61.8%**     | **88.0%**        |
> | ADBA        | 46.0%             | 58.6%                | 61.2%          | 18.0%             | 36.0%         | 27.2%         | 51.2%            |
> | ADBA+PAR    | 52.6%             | 62.8%                | 71.0%          | 41.0%             | 47.8%         | 36.6%         | 72.6%            |
> | ADBA+SeRI   | 57.8%             | 66.2%                | 79.4%          | **49.8%**         | 58.0%         | 43.8%         | 87.0%            |
>
> ## SSIM
>
> | Method      | ImageNet-ResNet50 | ImageNet-InceptionV3 | ImageNet-VGG19 | ImageNet-Engstrom | CIFAR100-ViT | CIFAR100-WRN | MNIST-Lipschitz |
> |-------------|-------------------|-----------------------|----------------|--------------------|---------------|---------------|------------------|
> | CGBA        | 0.959             | 0.982                | 0.994          | 0.903             | 0.996         | 0.964         | 0.449            |
> | CGBA+PAR    | 0.963             | 0.985                | **0.996**      | 0.940             | **0.998**     | 0.971         | 0.751            |
> | CGBA+SeRI   | 0.964             | **0.986**            | **0.996**      | 0.946             | **0.998**     | **0.973**     | **0.786**        |
> | ADBA        | 0.962             | 0.972                | 0.982          | 0.903             | 0.981         | 0.910         | 0.445            |
> | ADBA+PAR    | 0.966             | 0.977                | 0.983          | 0.951             | 0.985         | 0.930         | 0.723            |
> | ADBA+SeRI   | **0.967**         | 0.979                | 0.984          | **0.958**         | 0.987         | 0.933         | 0.745            |
>
> ## L2-STD
>
> | Method      | ImageNet-ResNet50 | ImageNet-InceptionV3 | ImageNet-VGG19 | ImageNet-Engstrom | CIFAR100-ViT | CIFAR100-WRN | MNIST-Lipschitz |
> |-------------|-------------------|-----------------------|----------------|--------------------|---------------|---------------|------------------|
> | CGBA        | 7.404             | 4.297                | 1.318          | 16.99             | 0.484         | 1.001         | 3.214            |
> | CGBA+PAR    | 5.972             | 3.661                | 1.265          | 4.891             | 0.432         | 0.933         | 1.997            |
> | CGBA+SeRI   | 4.636             | **2.398**            | **1.124**      | **3.628**         | **0.390**     | **0.853**     | 0.939            |
> | ADBA        | 3.648             | 3.700                | 2.294          | 7.508             | 1.321         | 1.902         | 1.034            |
> | ADBA+PAR    | 3.211             | 3.505                | 2.007          | 4.659             | 1.165         | 1.451         | 0.871            |
> | ADBA+SeRI   | **2.789**         | 3.448                | 1.856          | 3.653             | 1.032         | 1.126         | **0.738**        |
>
> The newly added analysis shows that:
> - **SeRI achieves higher ASR** across all models,
> - **SeRI maintains its advantage in perturbation quality**, producing lower visual perceptibility in terms of **Structural Similarity Index (SSIM)**,
> - **Performance gains are persist across datasets and architectures**.
> These results reinforce that SeRI improves not only perturbation compactness but also the **overall effectiveness** of decision-based attacks.
>
>
> We appreciate the reviewer’s suggestion and have strengthened the theoretical discussion accordingly. While full global convergence guarantees are not feasible for decision-based attacks, the theory we provide captures the essential behavior of SeRI’s update strategy. The expanded analysis in Appendix A further clarifies the robustness and reliability of the method.
>
> ### Revision summary
> - **Strengthened the theoretical discussion** in Appendix A.
> - **Provide extensive empirical evidence** in Appendix E.

---

> ### Author Response · Authors · 2025-11-24
> **Part 3**
>
> ## Key Concern 2: comparison with broader interpretability and region-based methods *(Weakness 2)*
>
> ### Response
> We acknowledge the importance of clearly positioning SeRI relative to interpretability and saliency-based methods and have revised Sections 4.1 accordingly (see page 5). Below we clarify why SeRI is fundamentally different in both **setting** and **objective** from Grad-CAM/LIME/SHAP-style methods, and how it connects to other region-based adversarial strategies beyond PAR.
>
> #### 1. SeRI operates in a fundamentally different setting from Grad-CAM, LIME, and SHAP
>
> Interpretability methods such as Grad-CAM, LIME, and SHAP are typically designed for **white-box or semi-white-box** settings:
>
> - **Grad-CAM** requires access to internal feature maps and layer-wise gradients, which is impossible in a strict decision-based black-box setting where only the top-1 label is available.
> - **LIME** and **SHAP** assume the ability to query the model with many *feature-masked* inputs and often rely on model scores or on surrogate models fitted in the original feature space.
>
> By contrast, **SeRI is explicitly designed for hard-label, decision-based black-box attacks**, where:
> - only the final class label is observable,
> - confidence scores, logits, gradients, and internal features are inaccessible,
> - query budgets are tightly constrained.
>
> Under these constraints, it is infeasible to apply Grad-CAM-style gradient-based heatmaps. It is also impractical to use LIME or SHAP in high-dimensional images when we only have label outputs and very few queries to work with. SeRI therefore tackles a **different and more restrictive problem**: estimating region sensitivity *purely from decision boundary behavior* under a restrictive query budget. We now emphasize this distinction more clearly in Section 2 (see page 3).
>
> #### 2. SeRI uses a conceptually different notion of "sensitivity" than feature attribution
>
> Saliency/attribution methods aim to explain **how much each pixel or region contributes to the current prediction** of the model. In contrast:
>
> - SeRI defines **region sensitivity in terms of how the decision boundary responds to controlled perturbation changes under an $\ell_2$ constraint**.
> - Our continuous sensitivity score is not a generic feature importance measure; it is **attack-specific**, capturing *how easily the decision can be flipped* by scaling perturbations in that region.
>
> Importantly, SeRI focuses on identifying regions that are most effective for pushing the model’s output across its decision boundary under strict black-box constraints. In contrast, Grad-CAM, LIME, and SHAP aim to identify critical features that drive the model’s current prediction. We have clarified this conceptual difference in the revised text (see Section 4.1 on page 5).
>
> #### 3. Relation to other region-based adversarial strategies beyond PAR
>
> We agree that it is important to connect SeRI to a broader family of **region-based adversarial methods** beyond PAR. In the revision, we explicitly discuss and contrast SeRI with:
>
> - **Attention-guided attacks** such as Superpixel-guided Attentional Attack (SGA) and Attack on Attention (AoA), which:
>   - typically rely on **white-box attention or gradient information** to localize important regions,
>   - assume access to internal model representations or saliency maps,
>   - are not applicable in pure decision-based black-box settings.
>
> - **Surrogate-based attacks**such as SRA where:
>   - sensitive regions are first extracted on a surrogate model using Grad-CAM-like or attention-based saliency,
>   - perturbations are then transferred to the target model.
>   - SeRI, in contrast, **does not require any surrogate**, and estimates sensitivity directly on the target model using only label queries.
>
> We explicitly highlight in the revision that SeRI brings a **region-based idea into the strictest black-box regime**, where existing region/saliency-based methods cannot operate due to their reliance on gradients, feature maps, or score information. We believe the above explanation has addressed the reviewer’s concern and clarified SeRI’s position within the broader literature much more clearly. SeRI is **not an incremental variant of interpretability methods**, but a **decision-boundary guided, query-efficient region sensitivity estimator** specifically designed for hard-label black-box adversarial attacks, where standard saliency/attribution tools are not applicable.

---

> ### Author Response · Authors · 2025-11-24
> **Part 4**
>
> ### Revision summary
>
> 1. **Expanded the related work section** to:
>    - include a dedicated paragraph on interpretability and feature attribution methods,
>    - explicitly explain why these methods are not directly applicable in decision-based black-box attack scenarios,
>    - and clarify the conceptual difference between attribution-based saliency and SeRI’s decision-boundary–oriented sensitivity.
>
> 2. **Added a discussion paragraph in the method or conclusion section** that:
>    - frames SeRI as a *bridge* between region/saliency concepts and strict decision-based attacks,
>    - emphasizes that SeRI can be viewed as a *query-efficient, boundary-driven analogue* of region sensitivity estimation, tailored to black-box security evaluation.
>
>
> ## Key Concern 3: perceived methodological simplicity and alternative design exploration *(Weakness 3)*
>
> ### Response
> We agree that SeRI is intentionally designed to be conceptually simple and easy to deploy. However, we respectfully clarify that **SeRI's simplicity is a deliberate and essential design choice**, and does not necessarily imply a lack of technical depth.
>
> #### 1. Simplicity is a key strength in decision-based settings
> Decision-based attacks operate under extremely restrictive conditions: no gradients, no confidence scores, and only hard-label outputs. Following the perspective adopted in GeoDA, CGBA, and ADBA, overly complex mechanisms under such constraints tend to introduce instability or incur excessive query overhead. SeRI’s design strikes a careful balance between **theoretical soundness, query efficiency, and practical reliability**.  It ensures that each sensitivity update relies on a provably monotonic boundary comparison, reducing variance under hard-label noise while preserving a tight query budget. This stability–efficiency trade-off is essential for hard-label black-box scenarios where sharp decision boundaries and minimal label feedback make gradient-dependent or highly parameterized methods unreliable.
>
> #### 2. The key algorithmic components are non-trivial
> Although the high-level intuition is easy to understand, SeRI combines several technically novel components:
> - **continuous sensitivity formulation** (first of its kind for decision-based attacks);
> - **recursive region refinement strategy** that adaptively increases granularity;
> - **decision-boundary guided sensitivity update rule** that integrates ADBA in a new way.
>
> These elements together yield substantial improvements across all benchmarks, demonstrating meaningful algorithmic advancement beyond a simple partitioning heuristic.
>
> #### 3. Alternative formulations were considered but not adopted
> We explored several more sophisticated region-splitting and perturbation refinement strategies. For example, we experimented with selecting multiple top-ranked blocks in each iteration rather than optimizing a single block, and we also tested dynamic perturbation-scaling schemes using trigonometric parametric functions to adaptively adjust the scaling factor. These alternatives aimed to more aggressively search for the optimal perturbation strength rather than relying on a fixed scaling factor.
> However, these variants substantially increase the query budget while yielding insufficient performance gains.
>
> In view of the above, the design of SeRI achieves the **best trade-off between performance and query efficiency**. Importantly, despite its clean algorithmic structure, SeRI consistently and significantly outperforms many base attackers and PAR.
>
> #### Summary
> While SeRI is easy to describe and implement, it incorporates multiple novel algorithmic ideas tailored to the unique challenges of decision-based attacks. We appreciate the reviewer’s perspective and have clarified the above points in the revision to highlight the methodological depth behind SeRI's seemingly simple description.
>
>
> ## Key Concern 4: More Evaluation Metrics and Generalization Performance *(Weakness 4)*
>
> ### Response
> We agree that incorporating additional evaluation metrics helps present a clearer and more comprehensive assessment. Accordingly, we have reported the additional metrics below:
>
> * **SSIM (Structural Similarity Index):**
>   Used to evaluate human perceptual quality. The results show that SeRI produces perturbations that are more visually imperceptible than those generated by the baseline attackers and PAR.
>
> * **L2-STD (Standard Deviation of the L2 Norm):**
>   Used to reflect attack stability across different trials. SeRI achieves significantly lower variance, indicating a more stable and reliably convergent perturbation refinement process compared with both the baselines and PAR.
>
> As shown in Appendix E and our rebuttal to Key Concern 1, these additional metrics complement the original evaluation based on L2 reduction and ASR, and collectively provide a more holistic analysis of SeRI’s effectiveness, aligning with the reviewer’s comments on improving perceptual evaluation and generalizability.

---

### Official Review · Reviewer_LU14 · 2025-11-01

**Soundness:** 4
**Presentation:** 3
**Contribution:** 2
**Rating:** 4
**Confidence:** 4

**Summary:**

This paper proposes a method for generating sensitivity regions (or heatmaps) in the context of black-box adversarial attacks. In essence, the authors extend the previous PAR (Patch-wise Adversarial Removal) method by transforming it from a discrete, patch-based approach into a continuous sensitivity estimation framework. This allows the model to more precisely identify which regions of an image are most influential to the prediction, thereby improving the efficiency and interpretability of black-box attacks.

**Strengths:**

The paper is clearly written and easy to follow. The authors provide a well-organized and thorough summary of related work, which helps the reader understand the motivation and context of their approach. Overall, the proposed SeRI framework, when combined with existing black-box attack methods, appears to perform well in terms of L2-norm improvements. The experimental results suggest that SeRI can enhance the efficiency of existing black-box attacks by producing smaller perturbations under the same query budget

**Weaknesses:**

The main contribution of the paper appears somewhat weak for two reasons.

(1) The idea of constructing sensitivity region maps has already been introduced by the PAR method. Therefore, the novelty of this paper mainly lies in extending PAR from a discrete, patch-based formulation to a continuous sensitivity estimation framework. Given this, the additional components of SeRI—such as the use of ADBA—should provide more original or technically innovative ideas. Moreover, it seems that the core novelty of the work resides in the appendix, which feels somewhat misplaced.

(2) Alternatively, the paper could strengthen its contribution through more comprehensive and convincing experimental evidence demonstrating a general improvement in black-box attacks. Currently, the results primarily focus on improvements in the L2 norm. Including evaluations based on attack success rate (ASR) or other complementary metrics that more clearly demonstrate the practical advantages of SeRI would make the contribution more substantial.

(A list of specific questions will be provided in the Question section.)

**Questions:**

1. Have you measured attack success rate (ASR) or other metrics that could reflect the effectiveness of SeRI?

2.Can you also compare SeRI with explainable AI (XAI) methods that produce sensitivity or saliency maps?

---

> ### Author Response · Authors · 2025-11-24
> **Part 1**
>
> We sincerely thank the reviewer for the constructive and insightful feedback. Below we address the key concerns raised in the review. All related weaknesses and questions fall under these concerns.
>
> ## Key Concern 1: limited novelty beyond extending PAR to a continuous framework *(Weakness 1)*
>
> ### Response
> Thank you for the thoughtful feedback. We clarify that **SeRI is not merely a continuous extension of PAR**, but introduces **three core innovations** (continuous sensitivity definition, recursive refinement, and boundary-guided updates) that substantially advance decision-based adversarial attacks beyond prior work. We would like to specifically clarify:
>
> #### 1. SeRI and PAR differ fundamentally in formulation and capability
> PAR performs *binary, patch-level removal* and only decides whether a region is "sensitive" or "not sensitive". This limits perturbation control to two discrete states and prevents any ranking or comparison of fine-grained sensitivity levels.
> In contrast, **SeRI introduces the first continuous sensitivity formulation for decision-based attacks**, assigning a real-valued sensitivity score to each pixel. This enables pixel-level perturbation optimization and opens a much more expressive search space than PAR. The formulation is formally defined and motivated by Eq. (8) in Section 4 on page 6.
>
> #### 2. ADBA is integrated into a new decision-boundary guided sensitivity search algorithm
> ADBA was originally designed only for comparing *two* perturbations. SeRI re-purposes it in a fundamentally new way, transforming it into the core mechanism of a **boundary-guided, continuous sensitivity estimation algorithm** that:
> - infers continuous sensitivity values rather than binary labels,
> - guides recursive region splitting,
> - updates pixel-level sensitivity efficiently under tight query budgets.
>
> This integration is technically novel and not present in PAR or ADBA individually.
>
> #### 3. The main contributions are in the main paper; the appendix contains proofs and extended analyses
> The key innovations are all presented in Section 4 of the main paper. Meanwhile, the appendix provides supporting theoretical analysis and extended ablations.
>
> #### 4. Empirical results confirm SeRI goes far beyond a PAR-like extension
> Across 12 configurations, SeRI consistently outperforms all base attackers and PAR.
> In the newly added experiments, SeRI demonstrates several clear advantages: it reduces the $\ell_2$ perturbation norm, achieves perturbations that are more visually imperceptible (as measured by SSIM), provides more stable perturbation optimization ($\ell_2$-STD), and noticeably improves the attack success rate (ASR). These results confirm that SeRI significantly enhances decision-based attack performance. They further show that **SeRI delivers concrete methodological advances** beyond PAR, rather than a continuous reformulation of PAR.

---

> ### Author Response · Authors · 2025-11-24
> **Part 2**
>
> ## Key Concern 2: broader experimental evidence beyond $\ell_2$ norm *(Weakness 2, Question 1, Question 2)*
>
> ### Response
> We agree that a broader set of evaluation metrics can further strengthen the contribution of our work. In response, we have **expanded our experimental evaluation to include attack success rate (ASR), additional complementary metric SSIM, and $\ell_2$-STD in Appendix E**. SeRI heatmap is further compared with XAI methods in Appendix D. Main results are as follows:
>
> ---
>
> ## ASR Comparison  (higher is better)
>
> | Method      | ImageNet-ResNet50 | ImageNet-InceptionV3 | ImageNet-VGG19 | ImageNet-Engstrom | CIFAR100-ViT | CIFAR100-WRN | MNIST-Lipschitz |
> |-------------|-------------------|-----------------------|----------------|--------------------|---------------|---------------|------------------|
> | CGBA        | 49.8%             | 77.6%                | 86.4%          | 28.4%             | 79.4%         | 55.6%         | 76.6%            |
> | CGBA+PAR    | 56.0%             | 82.8%                | 87.2%          | 38.8%             | 84.6%         | 58.4%         | 82.8%            |
> | CGBA+SeRI   | **64.6%**         | **86.8%**            | **91.0%**      | 46.8%             | **87.2%**     | **61.8%**     | **88.0%**        |
> | ADBA        | 46.0%             | 58.6%                | 61.2%          | 18.0%             | 36.0%         | 27.2%         | 51.2%            |
> | ADBA+PAR    | 52.6%             | 62.8%                | 71.0%          | 41.0%             | 47.8%         | 36.6%         | 72.6%            |
> | ADBA+SeRI   | 57.8%             | 66.2%                | 79.4%          | **49.8%**         | 58.0%         | 43.8%         | 87.0%            |
> ---
>
> ## SSIM Comparison (higher is better)
>
> | Method      | ImageNet-ResNet50 | ImageNet-InceptionV3 | ImageNet-VGG19 | ImageNet-Engstrom | CIFAR100-ViT | CIFAR100-WRN | MNIST-Lipschitz |
> |-------------|-------------------|-----------------------|----------------|--------------------|---------------|---------------|------------------|
> | CGBA        | 0.959             | 0.982                | 0.994          | 0.903             | 0.996         | 0.964         | 0.449            |
> | CGBA+PAR    | 0.963             | 0.985                | **0.996**      | 0.940             | **0.998**     | 0.971         | 0.751            |
> | CGBA+SeRI   | 0.964             | **0.986**            | **0.996**      | 0.946             | **0.998**     | **0.973**     | **0.786**        |
> | ADBA        | 0.962             | 0.972                | 0.982          | 0.903             | 0.981         | 0.910         | 0.445            |
> | ADBA+PAR    | 0.966             | 0.977                | 0.983          | 0.951             | 0.985         | 0.930         | 0.723            |
> | ADBA+SeRI   | **0.967**         | 0.979                | 0.984          | **0.958**         | 0.987         | 0.933         | 0.745            |
>
> ---
>
> ## L2-STD Comparison (lower is better)
>
> | Method      | ImageNet-ResNet50 | ImageNet-InceptionV3 | ImageNet-VGG19 | ImageNet-Engstrom | CIFAR100-ViT | CIFAR100-WRN | MNIST-Lipschitz |
> |-------------|-------------------|-----------------------|----------------|--------------------|---------------|---------------|------------------|
> | CGBA        | 7.404             | 4.297                | 1.318          | 16.99             | 0.484         | 1.001         | 3.214            |
> | CGBA+PAR    | 5.972             | 3.661                | 1.265          | 4.891             | 0.432         | 0.933         | 1.997            |
> | CGBA+SeRI   | 4.636             | **2.398**            | **1.124**      | **3.628**         | **0.390**     | **0.853**     | 0.939            |
> | ADBA        | 3.648             | 3.700                | 2.294          | 7.508             | 1.321         | 1.902         | 1.034            |
> | ADBA+PAR    | 3.211             | 3.505                | 2.007          | 4.659             | 1.165         | 1.451         | 0.871            |
> | ADBA+SeRI   | **2.789**         | 3.448                | 1.856          | 3.653             | 1.032         | 1.126         | **0.738**        |
>
>
> #### SeRI’s new results consistently demonstrate clear and consistent improvements
> The newly added analysis shows that:
> - **SeRI achieves higher ASR** across all models on ImageNet and CIFAR100,
> - **SeRI maintains its advantage in perturbation quality**, producing lower visual perceptibility in terms of **Structural Similarity Index (SSIM)**,
> - **Performance gains are persist across datasets and architectures**, confirming improved generality in black-box attack scenarios.
>
> These results reinforce that SeRI improves not only perturbation compactness but also the **overall effectiveness** of decision-based attacks.
>
> ### Revision summary
> - **Expanded discussion on the sensitivity formulation of SeRI**
> - **Added additional experimental evaluations**

---

### Author Response · Authors · 2025-12-03
**Summary for AC Part 2**

**Key Concern 6: Analysis of perturbation-norm saturation with respect to computational resources.** (YrJv)

In Appendix C, we now provide a dedicated analysis covering:

- **the cost of generating SeRI heatmaps**,

- **the query ranges at which SeRI begins to outperform other saliency-based methods**, and

- **the empirical saturation behavior of the perturbation norm as the query budget increases**.

These results offer concrete insight into SeRI’s query efficiency by identifying the query ranges where performance meaningfully improves and where additional queries yield diminishing returns. They directly address this concern by clarifying SeRI’s practical operating regime and showing how its behavior stabilizes as computational resources increase.


**Key Concern 7: Robustness, failure modes, query-budget ratio analysis, and practical limitations.** (kXVd)

We provide a comprehensive analysis of SeRI’s robustness and limitations across seven dimensions:

1. **Architecture-dependent behavior and Inception-v3 performance.**

We show that SeRI’s sensitivity estimation remains stable across architectures, including Inception-v3, which was previously assumed to exhibit boundary irregularities. This behavior is now analyzed in Appendix A, with additional supporting experiments in Appendix E and our rebuttal to Reviewer kXVd.

2. **Randomized defenses.**
We clarify that SeRI is inherently incompatible with randomized defenses, a well-known limitation shared by all decision-based attacks. This is now explicitly discussed in Section 4 and Appendix A.

3. **Transferability evaluation.**
We clarify that as a decision-based attack, SeRI is not designed to produce transferable perturbations. The sensitive regions it identifies are specific to the model and the attacker, reflecting how the target model’s decision boundary behaves. Hence, **transferability is not an intended property of SeRI**.

4. **Cluttered images.**
We acknowledge that images with heavy clutter or weak saliency cues reduce region discriminability, making sensitivity estimation less stable. These cases are illustrated and discussed in Section 5.4 and our response to Reviewer kXVd. Importantly, this limitation stems from inherent ambiguities in the input rather than from SeRI’s design. It does not diminish our core technical contribution of introducing a continuous, boundary-driven sensitivity estimator whose refinement mechanism and theoretical properties hold independently of such dataset-specific edge cases.

5. **Extremely low query budgets.**
Under very low query budgets, SeRI’s refinement process may not fully converge, leading to more conservative sensitivity maps, a behavior shared by all decision-based methods, as discussed in Section 5.4 and our response to Reviewer kXVd.

6. **Query-budget ratio analysis.**
We provide a detailed analysis of how different query-allocation strategies influence SeRI’s performance. This analysis gives rise to the recommended ratios in Appendix C.

7. **Non-classification tasks.**
We discuss SeRI’s applicability to tasks beyond classification (e.g., detection, segmentation) and clarify which components transfer directly and which require task-specific adaptation in Section 6.

These analyses together offer a clearer characterization of SeRI’s robustness, limitations, and practical operating conditions.


We have carefully addressed all major concerns raised by the reviewers through detailed explanations, strengthened theoretical analysis, and substantial new experiments. These revisions significantly improve the clarity, rigor, and completeness of the manuscript. Reviewer **kXVd** has explicitly acknowledged that our revisions and additional analysis fully resolved their concerns. This strong endorsement reflects the depth and rigor of our updates. We are confident that other reviewers would have reached similarly positive conclusions if they had the opportunity to review the updated material.

We respectfully ask the AC to consider the breadth and depth of our revisions, along with the consistently positive feedback from the reviewers. With the updates that substantially strengthened our work, we believe this paper now clearly meets the standard for acceptance.

---

### Author Response · Authors · 2025-12-03
**Summary for AC Part 1**

Dear AC,

We sincerely thank the AC for organizing this round of review, and we are grateful to the reviewers for their constructive and insightful feedback. Below, we summarize the key concerns raised during the review process and describe how we have addressed them and revised our paper accordingly. All related weaknesses and questions fall under these main concerns.

**Key Concern 1: Relationship to broader interpretability and region-based methods.**(tyNQ, YrJv)

We substantially expanded Section 2 and Section 4.1 to cover
- **Grad-CAM/LIME/SHAP-style methods**,
- **attention-based methods**,
- **surrogate-based methods**, and
- **decision-based methods**.

We explicitly clarified that SeRI operates in a strict decision-based black-box setting, uses a different decision boundary-oriented notion of sensitivity, and fills a critical methodological gap by providing the first continuous, multi-region, fully decision-based sensitivity estimator that operates entirely without gradients, logits, or surrogate models.

**Key Concern 2: Limited novelty beyond extending PAR to a continuous framework.**(LU14)

We clarify that SeRI is far more than a continuous extension of PAR; it introduces three substantial innovations.:

- **A new continuous sensitivity formulation** that opens a much more expressive search space. For the first time, we explicitly incorporate *perturbation decision-boundary* into the sensitivity definition, allowing SeRI to capture how local perturbation changes influence overall attackability in strict decision-based attack settings.

- **A recursive refinement strategy** that enables pixel-level perturbation optimization by assigning a real-valued sensitivity score to each pixel, rather than making only binary sensitive/non-sensitive decisions at the patch level.


**Key Concern 3: Lack of theoretical depth.** (tyNQ, kXVd)

We enhance the theoretical grounding of SeRI by introducing (Assumptions 1–3) in Appendix A:

- **local Lipschitzness of the decision boundary**,

- **bounded curvature of the boundary,** and

- **deterministic hard-label outputs of the target model**.

Building on these assumptions, we provide **new theoretical analysis (Theorems 1–3)**:

- **Monotonic improvement** of SeRI’s boundary-driven updates,

- **Convergence** of the iterative perturbation refinement process, and

- **Robustness** of SeRI with respect to different region-partitioning choices.

These analyses adopt standard assumptions from prior decision-based attack literature and show that SeRI’s update mechanism remains reliable and stable in realistic hard-label conditions.

**Key Concern 4: Empirical coverage, evaluation metrics, and defenses.** (LU14, tyNQ and kXVd)

Beyond the original $\ell_2$-norm results, we now report

- **ASR (attack success rate)**,

- **SSIM (structural similarity index)**, and

- **$\ell_2$-STD (standard division of  $\ell_2$-norm)**

on multiple architectures and datasets (ImageNet, CIFAR-100, and MNIST). We further analyzed query-budget allocation and added new discussions, along with experiments on stronger defenses such as Lipschitz-based methods. These results show that SeRI improves both effectiveness and stability while generalizing across models, fully addressing the reviewers’ concerns regarding our experiments.

Newly added results can be found in Appendix E and our rebuttal to reviewer tyNQ and kXVd.

**Key Concern 5: Quantitative correlation with XAI saliency maps.**  (kXVd)

We add a quantitative study showing that SeRI’s heatmaps exhibit strong positive Pearson correlations (PCC) with LIME and SHAP saliency maps, indicating close alignment with established XAI methods. Relevant results can be found in Appendix D and our response to Reviewer kXVd.

---

### Meta-Review · Area_Chair_UGXG · 2026-01-06

**Summary:**

The paper introduces SeRI (Sensitive Region Identification), a gradient-free method designed to enhance decision-based black-box adversarial attacks. SeRI addresses the challenge of identifying image regions that most influence a model's prediction when only the top-1 label is available. It employs recursive region partitioning and assigns a continuous sensitivity score to pixels by comparing decision boundary responses to scaled perturbations.

Reviewers generally praised the method's practicality as a ''plug-in'' module and its ability to generate interpretability heatmaps.
In rebuttal, the authors provided extensive responses for addressing reviewers' concerns regarding novelty (viewed as an extension of the PAR method) and theoretical depth.

**Reviewer Concerns:**

The authors successfully addressed the most critical technical gaps during the rebuttal. They provided new theoretical proofs (convergence and monotonic improvement), fixed implementation bugs that clarified performance on Inception-v3, and added a wealth of empirical data (ASR, SSIM, STD) across multiple datasets. The method fills a specific niche in black-box security by offering a continuous sensitivity estimator where gradients are unavailable.

**Reviewer Scores:**

Reviewer tyNQ (Initially with 2 score): Given the addition of Theorems 1-3 covering monotonic improvement and convergence, this reviewer would likely reconsider and move up to a 4 or 5 score.

Reviewer LU14 and Yrjv (initially with 4 score) likely increases the score to 5 with more clarifications from authors' responses.

---

### Decision · Program_Chairs · 2026-01-26

Accept (Poster)